# Tissue resident and follicular Treg cell differentiation is regulated by CRAC channels

Martin Vaeth [1,5], Yin-Hu Wang[1], Miriam Eckstein[2,5], Jun Yang[1], Gregg J. Silverman [3], Rodrigo S. Lacruz[2], Kasthuri Kannan[1,4] & Stefan Feske[1]

T regulatory (Treg) cells maintain immunological tolerance and organ homeostasis. Activated Treg cells differentiate into effector Treg subsets that acquire tissue-specific functions. $Ca^{2+}$ influx via $Ca^{2+}$ release-activated $Ca^{2+}$ (CRAC) channels formed by STIM and ORAI proteins is required for the thymic development of Treg cells, but its function in mature Treg cells remains unclear. Here we show that deletion of *Stim1* and *Stim2* genes in mature Treg cells abolishes $Ca^{2+}$ signaling and prevents their differentiation into follicular Treg and tissue-resident Treg cells. Transcriptional profiling of STIM1/STIM2-deficient Treg cells reveals that $Ca^{2+}$ signaling regulates transcription factors and signaling pathways that control the identity and effector differentiation of Treg cells. In the absence of STIM1/STIM2 in Treg cells, mice develop a broad spectrum of autoantibodies and fatal multiorgan inflammation. Our findings establish a critical role of CRAC channels in controlling lineage identity and effector functions of Treg cells.

[1] Department of Pathology, New York University School of Medicine, New York, NY 10016, USA. [2] Department of Basic Science and Craniofacial Biology, New York University College of Dentistry, New York, NY 10010, USA. [3] Department of Medicine, New York University School of Medicine, New York, NY 10016, USA. [4] Genome Technology Center, New York University School of Medicine, New York, NY 10016, USA. [5] Present address: Institute for Systems Immunology, Julius-Maximilians University of Würzburg, 97078 Würzburg, Germany. Correspondence and requests for materials should be addressed to S.F. (email: feskes01@nyumc.org)

T regulatory (Treg) cells are a subset of CD4[+] T cells with immunosuppressive function that are critical for immune homeostasis and the prevention of autoimmunity. Treg cells, which constitute ~5–15% of the peripheral T cell pool[1], are characterized by the expression of the transcription factor forkhead box P3 (Foxp3) and the high-affinity IL-2 receptor alpha chain (CD25). The importance of Foxp3 as the master regulator of Treg cells is evident from Scurfy mice and patients with immunodysregulation polyendocrinopathy enteropathy X-linked (IPEX) syndrome with loss-of-function mutations in *Foxp3* who suffer from multiorgan autoimmunity[2,3]. Nevertheless, Foxp3 alone is not sufficient for Treg differentiation and function as ectopic Foxp3 expression alone in CD4[+] T cells is unable to reproduce the transcriptional signature and function of Treg cells[4]. Furthermore, targeted deletion of *Foxp3* in mature Treg cells did not interfere with key characteristics of Treg cells, such as their anergic phenotype and expression of Treg markers (e.g. CD25, CTLA4, and Helios)[5]. These data suggest that additional signaling pathways are required for the identity and function of Treg cells, but the nature of these signals is incompletely understood.

Foxp3[+] Treg cells can be classified into thymus-derived (or natural) tTregs and peripherally induced pTregs that have complementary roles but differ significantly in their stability, antigen-specificity and regulatory function[1]. pTregs are derived from naïve conventional CD4[+] T cells that acquire transient Foxp3 expression after T cell receptor (TCR) stimulation in the presence of transforming growth factor beta (TGFβ) and/or the absence of co-stimulatory signals. By contrast, tTregs represent a stable T cell lineage that develops during thymic negative selection and exhibits a unique transcriptional and epigenetic program that is critical for their sustained regulatory function[1,6]. Upon activation, tTreg cells can further differentiate into specialized effector Treg subsets, such as tissue-resident, memory-like Treg cells that have important roles in the function of non-lymphoid organs[6,7], as well as T follicular regulatory (Tfr) cells that shape the quality and quantity of humoral immune responses during the germinal center (GC) reaction[8–10]. These effector Treg cells differ significantly from Treg cells in secondary lymphoid organs because they acquire a tissue-specific gene expression program that includes transcription factors, homing receptors, and tissue-adapted regulatory molecules, which are not or only weakly expressed in lymphoid tissue Treg cells[6,7].

How this functional specification occurs is not well understood but it is believed that tissue-specific cues induce a gene expression program that co-opts the surrounding tissue, and promotes site-specific functions of Treg cells[6]. Distinct populations of Treg cells with organ-specific functions have been identified in many non-lymphoid tissues including small and large intestine, skin, lung, liver, adipose tissue, skeletal muscle, and various tumors. Skin-resident Treg cells express the transcription factor RORα and promote immune tolerance to skin commensals, wound healing, and hair follicle regeneration[11–13]. In skeletal muscle, a small but distinct population of Treg cells expands rapidly after muscle injury and promotes myocyte regeneration through expression of the growth factor Amphiregulin[14]. In visceral adipose tissue (VAT), Treg cells express the adipose tissue-specific transcription factor peroxisome proliferator-activated receptor gamma (PPARγ) and modulate the insulin sensitivity of adipocytes[15]. Similar to tissue-resident Treg cells, Tfr cells are effector Treg cells that co-opt the transcriptional program of their lymph follicle environment. Like T follicular helper (Tfh) cells, Tfr cells express CXCR5, PD-1, ICOS, and the transcription factor Bcl-6[8,9]. In contrast to Tfh cells, Tfr cells lack molecules that provide B cell help, such as CD40L, IL-21, and IL-4, but instead express regulatory molecules like IL-10, CTLA-4 and the

transcriptional regulator Blimp-1 (encoded by *Prdm1*). Tfr cells promote the selection of high-affinity B cell clones and prevent uncontrolled GC reactions and humoral autoimmunity[8–10].

Stimulation of Treg cells through their TCR triggers the activation of phospholipase Cγ1 (PLCγ1) and the production of inositol-1,4,5-trisphosphate (IP$_3$). IP$_3$ binds to and opens IP$_3$ receptor channels in the endoplasmic reticulum (ER) resulting in the depletion of calcium (Ca$^{2+}$) from ER stores and a transient increase in intracellular Ca$^{2+}$ levels[16]. Ca$^{2+}$ depletion from the ER also causes the activation of two proteins in the ER membrane, stromal interaction molecule 1 (STIM1) and STIM2, allowing them to bind to ORAI proteins in the plasma membrane. ORAI1 and its homologs ORAI2 and ORAI3 form the Ca$^{2+}$ release-activated Ca$^{2+}$ (CRAC) channel[17,18] that mediates robust and sustained Ca$^{2+}$ influx in T cells[19]. This form of Ca$^{2+}$ influx is called store-operated Ca$^{2+}$ entry (SOCE). The resulting Ca$^{2+}$ signaling leads to the activation of multiple Ca$^{2+}$-regulated transcription factors, such as the nuclear factor of activated T cells (NFAT), nuclear factor kappa-light-chain enhancer of activated B cells (NF-κB), cAMP response element-binding protein (CREB), and others[20].

SOCE is required for the development of Foxp3[+] Treg cells in the thymus. Mice lacking *Stim1* and *Stim2* in T cells have reduced tTreg numbers in the thymus and secondary lymphoid organs, which was partly due to impaired IL-2 signaling in SOCE-deficient Treg cells[21,22]. The molecular mechanisms how SOCE regulates *Foxp3* expression and Treg development remain largely unclear. In pTreg cells, SOCE controls *Foxp3* expression through NFAT binding to the conserved non-coding DNA sequence 1 (CNS1) within the *Foxp3* gene locus, and *Stim1*-deficient CD4[+] T cells are impaired in their ability to differentiate into pTreg cells[23]. CNS1 is not required, however, for the development of tTreg cells[24,25] and Foxp3 expression in tTreg cells appears to be less dependent on strong NFAT activation.

The importance of SOCE in T cells for peripheral immune tolerance is emphasized by patients with loss-of-function mutations in *STIM1* and *ORAI1* genes, who suffer from immunodeficiency and autoimmunity including autoimmune hemolytic anemia (AIHA)[26]. Several patients had reduced numbers of Foxp3[+] Treg cells in their blood[27,28] consistent with a role of SOCE in Treg development, whereas others had normal Treg numbers despite symptoms of autoimmunity[29,30]. The latter finding suggests that SOCE has additional roles in Treg cells which may include controlling Treg function or the differentiation into effector Treg cells, such as Tfr and tissue-resident Treg cells. We had shown that deletion of SOCE in pTreg[23] and tTreg[21,22] cells impairs their suppressive function, but the mechanisms by which SOCE regulates Treg function remain unclear.

Using mice lacking *Stim1* and *Stim2* selectively in mature Foxp3[+] Treg cells, we here show that SOCE is critical for the differentiation and function of effector Treg cells. Although these mice have largely normal Foxp3[+] Treg cell numbers in their lymphoid organs, they develop an early onset, type 2 autoimmune disease that is dominated by autoantibodies and severe multi-organ inflammation. This phenotype is associated with a strong reduction of tissue-resident Treg cell numbers in multiple organs and Tfr cells in secondary lymphoid organs. We show that STIM1/STIM2 and SOCE control a complex transcriptional network that coordinates the expression of genes regulating cell cycle, metabolism, cytokine signaling, and Treg effector functions. Taken together, we report that Ca$^{2+}$ signaling downstream of CRAC channel activation is an essential pathway for the differentiation of Treg cells into Tfr and tissue-resident Treg cells. These findings indicate that CRAC channels have dual and separable roles in the thymic development of Treg cells, and in their differentiation into effector Treg subsets in the periphery.

## Results

**STIM1/STIM2 in Treg cells prevent fatal systemic inflammation**. To investigate the role of SOCE in Treg cells after their development in the thymus, we generated $Stim1^{fl/fl}Stim2^{fl/fl}$ $Foxp3$-YFPcre mice (abbreviated $Stim1/2^{Foxp3}$) to delete both $Stim1$ and $Stim2$ genes conditionally in mature Foxp3$^+$ Treg cells. Because expression of $Cre$ recombinase is under the control of the $Foxp3$ promoter on the X chromosome, all Tregs in hemizygous male $Stim1/2^{Foxp3}$ mice are expected to lack $Stim1$ and $Stim2$. We confirmed the successful ablation of $Stim1$ and $Stim2$ expression in Foxp3$^+$ Treg cells isolated from male $Stim1/2^{Foxp3}$ mice by RT-PCR (Fig. 1a). Measurements of SOCE in Treg cells after ER store depletion with the sarco/endoplasmic $Ca^{2+}$ ATPase (SERCA) inhibitor thapsigargin (TG) confirmed that SOCE was abolished in STIM1/2-deficient Treg cells (Fig. 1b). It is noteworthy that SOCE was lower in Treg than conventional T cells from WT mice (Fig. 1b), which may be due to higher $Stim2$ but lower $Stim1$ expression in Treg cells (Fig. 1a). These findings are consistent with previous observations in murine Treg cells and the less efficient activation of SOCE by STIM2 compared to STIM1[24,31,32]. Male $Stim1/2^{Foxp3}$ mice had normal or elevated numbers of Foxp3$^+$ Treg cells in their thymus and secondary lymphoid organs compared to WT littermates (Fig. 1c), despite the fact that their Foxp3 mRNA and protein levels were slightly reduced (Fig. 1a, Supplementary Figure 1a, b). These findings demonstrate that deletion of STIM1 and STIM2 in mature tTregs by Foxp3-Cre does not interfere with the maintenance of Treg cells.

tTregs can be characterized as naïve (CD62L$^{hi}$CD44$^{lo}$) or activated (CD62L$^{lo}$CD44$^{hi}$), the latter being maintained by continuous TCR stimulation[33,34]. Male $Stim1/2^{Foxp3}$ mice showed a similar distribution of naive and activated Foxp3$^+$ Treg cells compared to WT littermate controls (Fig. 1d) suggesting that their activation by self-antigens and maintenance is independent of SOCE. Despite normal numbers and activation of tTreg cells in the absence of SOCE, male $Stim1/2^{Foxp3}$ mice developed a severe disease phenotype with a hunched posture, scaly skin, and alopecia starting around 4 weeks of age (Fig. 1e). Male $Stim1/2^{Foxp3}$ mice were born at a normal Mendelian ratio, but failed to thrive and rapidly lost weight after weaning compared to WT littermate controls (Fig. 1f), and died prematurely <8 weeks post partum (Fig. 1g). Collectively, these findings suggested that while SOCE in Treg cells is dispensable for the maintenance of Treg cell numbers, it is critical for the health and survival of mice.

**SOCE in Treg cells prevents Th2-mediated autoimmunity**. Dissection of male $Stim1/2^{Foxp3}$ mice revealed splenomegaly and lymphadenopathy with significantly enlarged submandibular, inguinal, and axillary LNs compared to WT littermate controls, whereas mesenteric LNs were less enlarged (Fig. 2a, Supplementary Figure 1c, d). Although the frequencies of conventional CD4$^+$ and CD8$^+$ T cells in the spleen, LNs, and blood were comparable between WT and $Stim1/2^{Foxp3}$ mice (Supplementary Figure 1e, f), their phenotype was dramatically skewed towards CD44$^{hi}$CD62$^{lo}$ effector T cells (Fig. 2b). Splenic CD4$^+$ T cells of $Stim1/2^{Foxp3}$ mice also showed higher Ki-67 expression compared to WT control mice (Fig. 2c) suggesting that enhanced T cell proliferation contributes to the splenomegaly and lymphadenopathy of $Stim1/2^{Foxp3}$ mice (Fig. 2a, Supplementary Figure 1c, d). To further characterize the phenotype of CD4$^+$ effector T cells in $Stim1/2^{Foxp3}$ mice, we measured the expression of Th1, Th2, and Th17 lineage-determining transcription factors. GATA3 expression was significantly higher in CD4$^+$ T cells isolated from the spleen and LNs of $Stim1/2^{Foxp3}$ mice compared to WT controls (Fig. 2c), whereas no differences in T-bet or RORγt expression

were observed (Supplementary Figure 1g) indicating that the immunopathology in $Stim1/2^{Foxp3}$ mice is driven by a type 2 immune response. This notion was further supported by the fact that the Th2 cytokines IL-4 and IL-5 were strongly increased in the sera of $Stim1/2^{Foxp3}$ mice, whereas levels of the Th1 and Th17 cytokines IFNγ and IL-17 were only moderately elevated (Fig. 2d). Consistent with increased Th2 cytokines, we observed significantly higher immunoglobulin (Ig) levels in the sera of $Stim1/2^{Foxp3}$ mice, in particular IgM and IgE (Fig. 2e). As a consequence, CD11b$^+$Siglec F$^+$ eosinophils were significantly elevated in the spleens of $Stim1/2^{Foxp3}$ mice (Fig. 2f). Collectively these data suggest that SOCE in Treg cells is required to prevent Th2-mediated immunopathology.

In contrast to male $Stim1/2^{Foxp3}$ mice, female heterozygous $Stim1/2^{Foxp3}$ mice were healthy and developed normally. They have a mixture of both WT (YFPcre$^−$) and $Stim1/2$-deficient (YFPcre$^+$) Treg cells (Supplementary Figure 2a). Their YFP$^+$ $Stim1/2$-deficient Treg cells lacked SOCE compared to YFP$^−$ WT Treg cells (Supplementary Figure 2b) and had slightly lower Foxp3 expression (Supplementary Figure 2a). To test the immunosuppressive function of $Stim1/2$-deficient Treg cells, we isolated CD4$^+$CD25$^{hi}$YFPcre$^−$ and CD4$^+$CD25$^{hi}$YFPcre$^+$ Treg cells from female $Stim1/2^{Foxp3}$ mice. The use of $Stim1/2$-deficient Treg cells from healthy female mice circumvents potential secondary effects on Treg function due to the inflammation in male $Stim1/2^{Foxp3}$ mice. To determine the suppressive function of WT and $Stim1/2$-deficient Treg cells in vitro, they were co-incubated with CFSE-labeled effector T cells. Although $Stim1/2$-deficient Treg cells had some suppressive effect on the proliferation of activated CD4$^+$ and CD8$^+$ T cells, their function was markedly impaired compared to WT Treg cells (Supplementary Figure 2c). To test the suppressive function of $Stim1/2$-deficient Treg cells in vivo, we co-transferred CD45.1$^+$ CD4$^+$CD25$^−$CD62L$^{hi}$ naive WT T cells together with CD45.2$^+$ WT or $Stim1/2$-deficient Treg cells into lymphocyte-deficient $Rag1^{−/−}$ mice (Supplementary Figure 2d). In this model, functional Treg cells prevent the development of inflammatory bowel disease (IBD) characterized by the differentiation of naive CD4$^+$ T cells into Th1 and Th17 cells and lymphocytic inflammation of the large intestine. We found that host mice receiving $Stim1/2$-deficient Treg cells had significantly reduced body weights 10–12 weeks after T cell transfer compared to mice receiving WT Treg cells (Fig. 2g) suggesting that SOCE in Treg cells is required to prevent IBD. Indeed, $Stim1/2$-deficient Treg cells failed to suppress the expansion of conventional T cells in spleens and mesenteric LNs (Fig. 2h) and their differentiation into Th1 and Th17 cells as indicated by increased IFNγ and IL-17 production, respectively, compared to WT Treg cells (Fig. 2i). Colon histologies showed immune cell infiltration, epithelial hyperplasia and goblet cell depletion in $Rag1^{−/−}$ mice 12 weeks after co-transfer of $Stim1/2$-deficient Treg cells, whereas host mice co-transplanted with WT Treg cells showed no obvious signs of IBD (Supplementary Figure 2e). Together, these data demonstrate that SOCE is required for the suppressive function of effector Treg cells.

**CRAC channels control transcriptional identity of Treg cells**. To understand at the molecular level how SOCE controls the immunosuppressive function of Treg cells, we performed transcriptional profiling of WT and $Stim1/2$-deficient Treg cells (Fig. 3 and Supplementary Figure 3). CD4$^+$CD25$^{hi}$YFPcre$^−$ (WT) and CD4$^+$CD25$^{hi}$YFPcre$^+$ ($Stim1/2$-deficient) Treg cells were isolated from healthy female heterozygous $Stim1/2^{Foxp3}$ mice (as shown in Supplementary Figure 2a), left untreated or stimulated for 16 h with anti-CD3/CD28 in vitro and analyzed

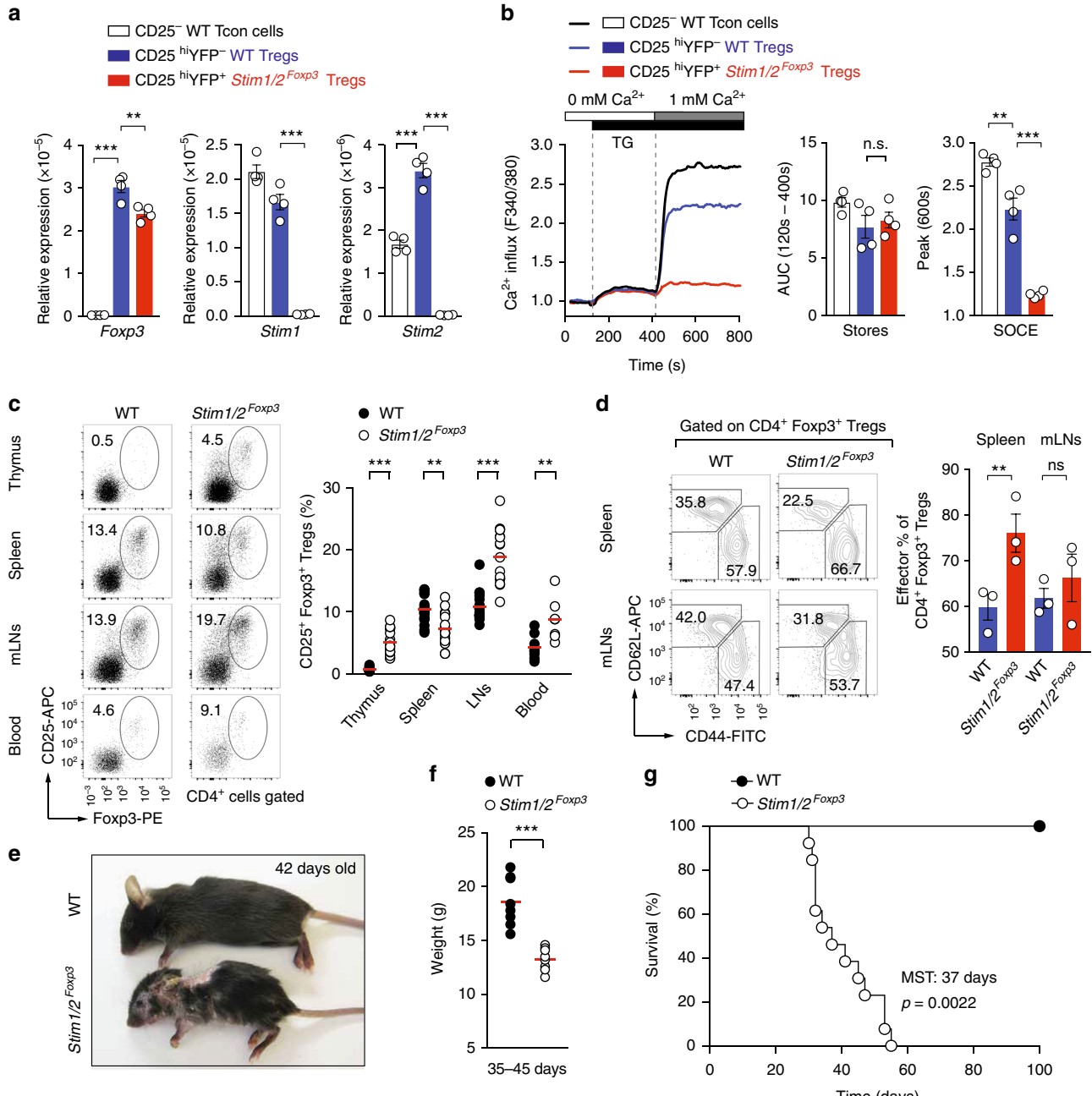

**Fig. 1** Deletion of STIM1 and STIM2 in Treg cells is fatal despite normal Treg numbers in secondary lymphoid organs. **a** Analysis of *Stim1*, *Stim2*, and *Foxp3* gene expression in conventional and Treg cells isolated from male WT and *Stim1*fl/fl*Stim2*fl/fl *Foxp3cre* (*Stim1/2*Foxp3) mice using qRT-PCR. Means ± SEM of four mice. **b** Analysis of $Ca^{2+}$ influx in T conventional (Tcon) and Treg cells isolated from male WT and *Stim1/2*Foxp3 mice loaded with Fura-2 and stimulated with thapsigargin (TG) in $Ca^{2+}$ free Ringer solution followed by addition of 1 mM $Ca^{2+}$ (left panel). Normalized store depletion ($AUC_{120s-400s}$) and SOCE ($peak_{600s}$) of T cells calculated as F340/380 emission ratios; means ± SEM of four mice (right panels). **c** Analysis of Foxp3+ Treg cells in secondary lymphoid organs and blood of male WT and *Stim1/2*Foxp3 mice using flow cytometry; means of 7–21 mice. **d** Analysis of naïve and effector Treg cells in spleen and mLNs of male WT and *Stim1/2*Foxp3 mice using flow cytometry. Bar graphs represent the means ± SEM of three mice. **e** Representative pictures of 42 days old male WT and *Stim1/2*Foxp3 mice. **f** Weight of 35–45 days old male WT and *Stim1/2*Foxp3 mice; means of eight mice. **g** Cumulative survival of male WT and *Stim1/2*Foxp3 mice; 9–13 mice per cohort. Mean survival time (MST) of *Stim1/2*Foxp3 mice 37 days and of WT mice > 100 days. Each dot in **c** and **f** represents one mouse. Statistical analysis in **a**–**d**, **f** by unpaired Student's *t*-test; in **g** using the Mantel–Cox test. *$p < 0.05$; **$p < 0.01$; ***$p < 0.001$. AUC, area under the curve

by RNA-sequencing (Fig. 3a). The gene expression profiles of unstimulated and anti-CD3/CD28-stimulated WT and *Stim1/2*-deficient Treg cells were clearly distinct (Supplementary Figure 3a). Of 1238 differentially expressed genes (DEG) in non-stimulated WT and *Stim1/2*-deficient Treg cells, more genes were downregulated than upregulated (771 vs. 467) >two-fold

(Fig. 3a, Supplementary Figure 3b, c). Similarly, in activated Treg cells more DEG were downregulated than upregulated (439 vs. 218) (Fig. 3a, Supplementary Figure 3b, c). We performed unbiased pathway enrichment and network analyses of DEG in unstimulated Treg cells to determine biological processes that are regulated by SOCE in Treg cells (Fig. 3b).

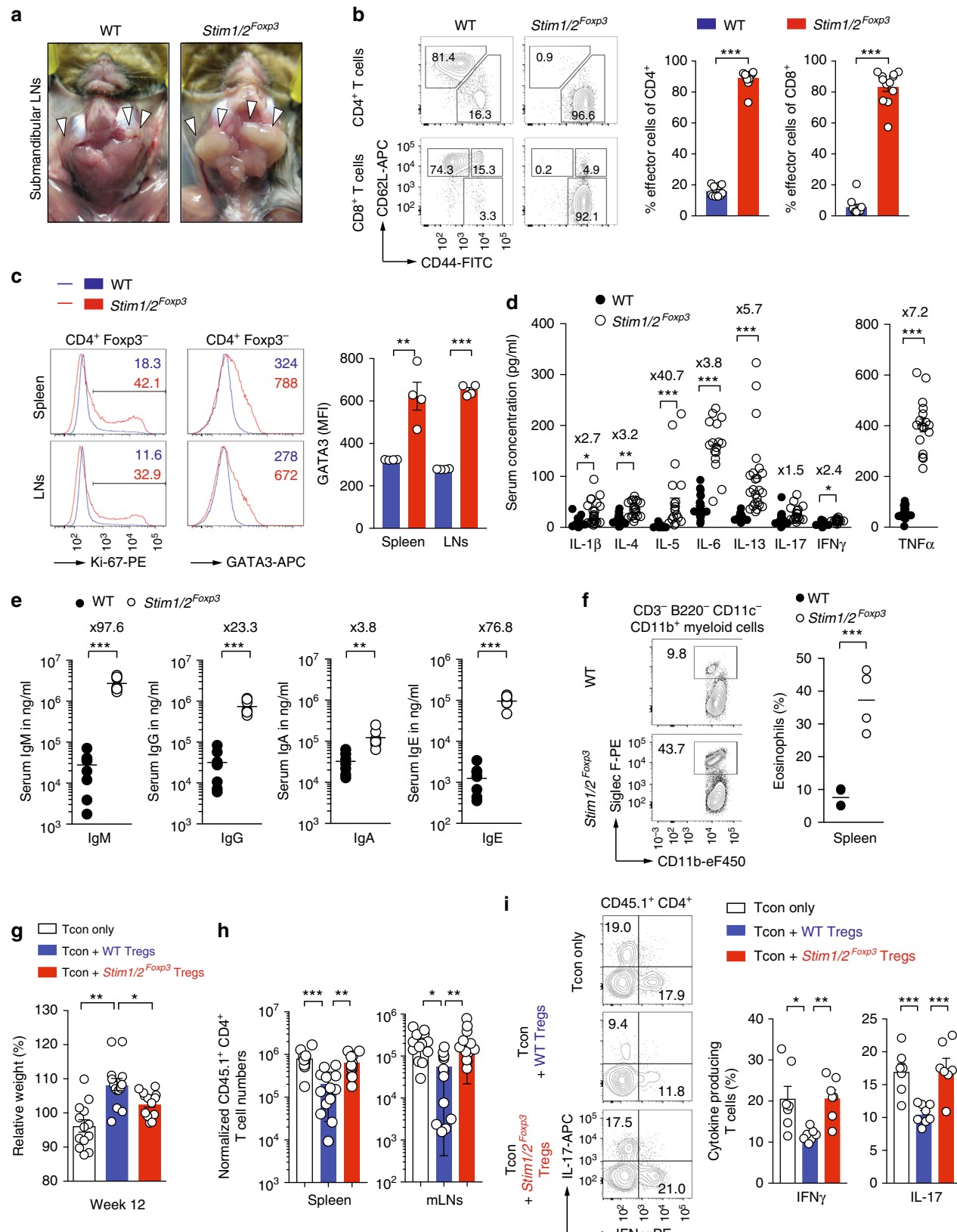

Although both upregulated and downregulated DEG were included in the analysis, the vast majority of pathways were negatively affected by the loss of SOCE in Tregs with the exception of one functional cluster (*humoral immune responses*) (Fig. 3b). The downregulated pathways were dominated by

*DNA replication*, *chromosomal organization* and *cell cycle regulation*, but also included various metabolic and immunological processes, such as *mitochondrial respiration*, *nucleotide metabolism*, and *adaptive immune responses* (Fig. 3b, Supplementary Figure 4a).

**Fig. 2** STIM1 and STIM2 control Treg function and prevent Th2-mediated autoimmunity. **a** Macroscopic analysis of submandibular LNs (arrows) in male WT and $Stim1/2^{Foxp3}$ mice. **b** Analysis of naive (CD62L$^{hi}$CD44$^{lo}$), effector (CD62L$^{lo}$CD44$^{hi}$), and memory (CD62L$^{hi}$CD44$^{hi}$) CD4$^+$ and CD8$^+$ T cells in the spleen of WT and $Stim1/2^{Foxp3}$ mice by flow cytometry; means ± SEM of 10–12 mice. **c** Analysis of cell cycle (Ki-67) and GATA3 expression in conventional CD4$^+$ T cells in the spleen and LNs of male WT and $Stim1/2^{Foxp3}$ mice by flow cytometry; means ± SEM of four mice. **d** ELISA measurements of serum cytokines in male WT and $Stim1/2^{Foxp3}$ mice; fold increases in $Stim1/2^{Foxp3}$ mice compared to WT mice are indicated by ×; means ± SEM of 13–20 mice. **e** Analysis of serum immunoglobulin concentrations in male WT and $Stim1/2^{Foxp3}$ mice using ELISA; means of eight mice. **f** Analysis of CD11b$^+$Siglec F$^+$ eosinophils in the spleen of male WT and $Stim1/2^{Foxp3}$ mice by flow cytometry; means of four mice. **g–i** $Stim1/2$-deficient Treg cells fail to suppress adoptive transfer colitis in lymphopenic host mice; for details see Supplementary Figure 2d. **g** Weight of $Rag1^{-/-}$ host mice 12 weeks after transfer of CD45.1$^+$ conventional CD4$^+$ WT T cells together with CD45.2$^+$ WT or $Stim1/2$-deficient Treg cells isolated from female heterozygous $Stim1/2^{Foxp3}$ mice; means ± SEM of 13–14 host mice. **h** Normalized numbers of CD45.1$^+$ conventional CD4$^+$ T cells in spleen and mLNs of $Rag1^{-/-}$ host mice 12 weeks after transfer; means ± SEM of 7–8 host mice. **i** Analysis of IL-17A and IFNγ production by CD45.1$^+$ conventional T cells from mLNs after PMA/ionomycin re-stimulation using flow cytometry; means ± SEM of 7–8 host mice. Statistical analysis in **b–g** and **i** by unpaired Student's $t$-test, in (**h**) by one-way ANOVA. *$p < 0.05$; **$p < 0.01$; ***$p < 0.001$

Among the most suppressed gene sets in the absence of SOCE was the *IL-2/STAT5 signaling* pathway (Fig. 3c, Supplementary Figure 4b)[35]. 36% of all genes in this pathway were down-regulated in $Stim1/2$-defcient Treg cells compared to 2% that were upregulated, suggesting that IL-2 signaling is strongly dependent on SOCE. This is intriguing because Treg development and function requires IL-2[36,37]. Besides canonical signaling through STAT5, IL-2 activates phosphatidylinositol 3-OH kinase (PI3K) upstream of the metabolic master regulator mechanistic target of rapamycin (mTOR). Although Treg cells express high levels of PTEN that antagonizes PI3K activity[38], mTOR is constitutively active in Treg cells[39]. The *mTORC1 signaling gene expression signature* was muted in $Stim1/2$-deficient Treg cells compared to controls (Supplementary Figure 4b, c) consistent with the notion that IL-2 sensitivity is reduced. IL-2 and mTOR signaling control the proliferation and function of Treg cells, at least in part, by regulating glycolysis, mitochondrial respiration and fatty acid oxidation (FAO)[38–40]. We found that the proliferation of $Stim1/2$-deficient Treg cells in vivo was decreased compared to their WT counterparts (Supplementary Figure 4d), which correlated with impaired expression of several metabolic pathways including glycolysis, OXPHOS, and TCA cycle (Supplementary Figure 4e). These data indicate that SOCE controls Treg metabolism through IL-2 signaling and mTOR activation.

Because IL-2 and mTOR signaling is critical for the maintenance, differentiation, and function of Treg cells[36,39], we compared DEGs in WT and $Stim1/2$-deficient Treg cells to the *common Treg gene expression signature*[4] that represents their transcriptional identity and distinguishes Treg from conventional T cells (Fig. 3d). More than 25% of all Treg-specific genes were deregulated in the absence of STIM1/STIM2 with the vast majority of transcripts being decreased (Fig. 3d). Among those genes were key markers of Treg cells, such as *Foxp3*, *Ctla4*, *Itgb8* (integrin β8), *Itgae* (integrin α$_E$), and *Tnfrsf9* (4-1BB) (Fig. 3d) suggesting that SOCE is required to maintain Treg identity. In addition, expression of numerous molecules with regulatory function in Treg cells including *Il10*, *Tgfb1*, *Ebi3* (IL-35), *Entpd1* (CD39), and *Areg* (Amphiregulin), as well as cell surface receptors, such as *Tnfrsf4* (Ox40), *Tnfrsf18* (GITR), *Pdcd1* (PD-1), *Icos*, *Tigit*, *Havcr2* (Tim-3), and *Nrp1*, was impaired in $Stim1/2$-deficient Treg cells compared to WT controls (Fig. 3e, g). Furthermore, expression of many transcription factors that control the differentiation and function of effector Treg cells was dependent on STIM1/STIM2 (Fig. 3f). For instance, *Irf4*, *Batf*, *Prdm1* (Blimp-1), *Ikzf2* (Helios), *Maf*, *Gata3*, *Ahr*, *Rora*, and *Rel* (p65/Rel-A) were downregulated, whereas expression of *Tcf7* (TCF-1) and *Klf2*, which antagonize Treg effector differentiation, were increased in $Stim1/2$-deficient Treg cells (Fig. 3f). Collectively, our data demonstrate that SOCE controls the identity of Treg cells by regulating the expression of transcription factors and molecules

that are critical for the function and differentiation of effector Treg cells.

**SOCE controls Tfr cell differentiation and autoimmunity.** Among the most significantly downregulated genes in $Stim1/2$-deficient Treg cells were *Pdcd1* (PD-1), *Icos*, and *Prdm1* (Blimp-1) (Fig. 4a). This was intriguing because these molecules are commonly used to identify Tfr cells. We therefore compared all DEGs in WT and $Stim1/2$-deficient Treg cells to the *Tfr cell gene expression signature* defined previously[8]. We observed that more than 35% of all Tfr-specific genes were deregulated in $Stim1/2$-deficient Treg cells with the vast majority being decreased (Fig. 4a). This finding suggested that SOCE is crucial for the differentiation of Tfr cells. Indeed, CD44$^+$CXCR5$^{hi}$PD-1$^{hi}$ Tfr cells were virtually absent in the secondary lymphoid organs of male $Stim1/2^{Foxp3}$ mice compared to WT littermate controls (Fig. 4b). As a consequence, male $Stim1/2^{Foxp3}$ mice showed spontaneous formation of PNA$^+$ GCs in their cervical LNs (Fig. 4c) and a marked accumulation of CD38$^-$GL.7$^+$ GC B cells in their spleens and LNs (Fig. 4d). Consistent with spontaneous GC formation (Fig. 4c, d) and elevated serum Ig levels (Fig. 2e), we found a broad spectrum of autoantibodies in the sera of male $Stim1/2^{Foxp3}$ mice, including antibodies against RNP/Sm, Ro/SSA, and Proteinase 3, which were absent in WT littermate controls (Fig. 4e). The sera of $Stim1/2^{Foxp3}$ mice also tested positive for antinuclear autoantibodies (ANA) and autoantibodies with a fine fibrous cytosolic staining pattern consistent with anti-mitochondrial antibodies (AMA) (Supplementary Figure 5a). Together these findings suggest that SOCE is required for the differentiation of Tfr cells and their ability to prevent spontaneous, likely autoantigen-driven, B cell maturation, and the production of autoantibodies.

Most patients with inherited mutations in *STIM1* and *ORAI1* that abolish SOCE develop humoral autoimmunity, which includes autoantibodies against red blood cells (RBCs) and AIHA (Supplementary Table 1)[26]. We therefore tested if $Stim1/2^{Foxp3}$ mice develop anti-RBC autoantibodies and AIHA (Fig. 4f–i). The sera of male $Stim1/2^{Foxp3}$ mice contained high titers of both IgM and IgG anti-RBC autoantibodies (Fig. 4f) and showed significantly increased IgG binding to Ter119$^+$ erythrocytes (Fig. 4g). Furthermore, peripheral blood smears revealed abnormal clumping and agglutination of $Stim1/2$-deficient RBCs and pathological alterations in their size and shape compared to RBCs of WT mice (Fig. 4h) indicative of AIHA. With increasing age, male $Stim1/2^{Foxp3}$ mice became progressively anemic with low RBC counts and hemoglobin levels (Fig. 4i, Supplementary Figure 5b). To demonstrate the pathogenicity of anti-RBC autoantibodies, we transferred serum from male WT or $Stim1/2^{Foxp3}$ mice into WT recipient mice. Serum from $Stim1/2^{Foxp3}$ (but not WT) mice caused a significant reduction of RBCs and hemoglobin levels in

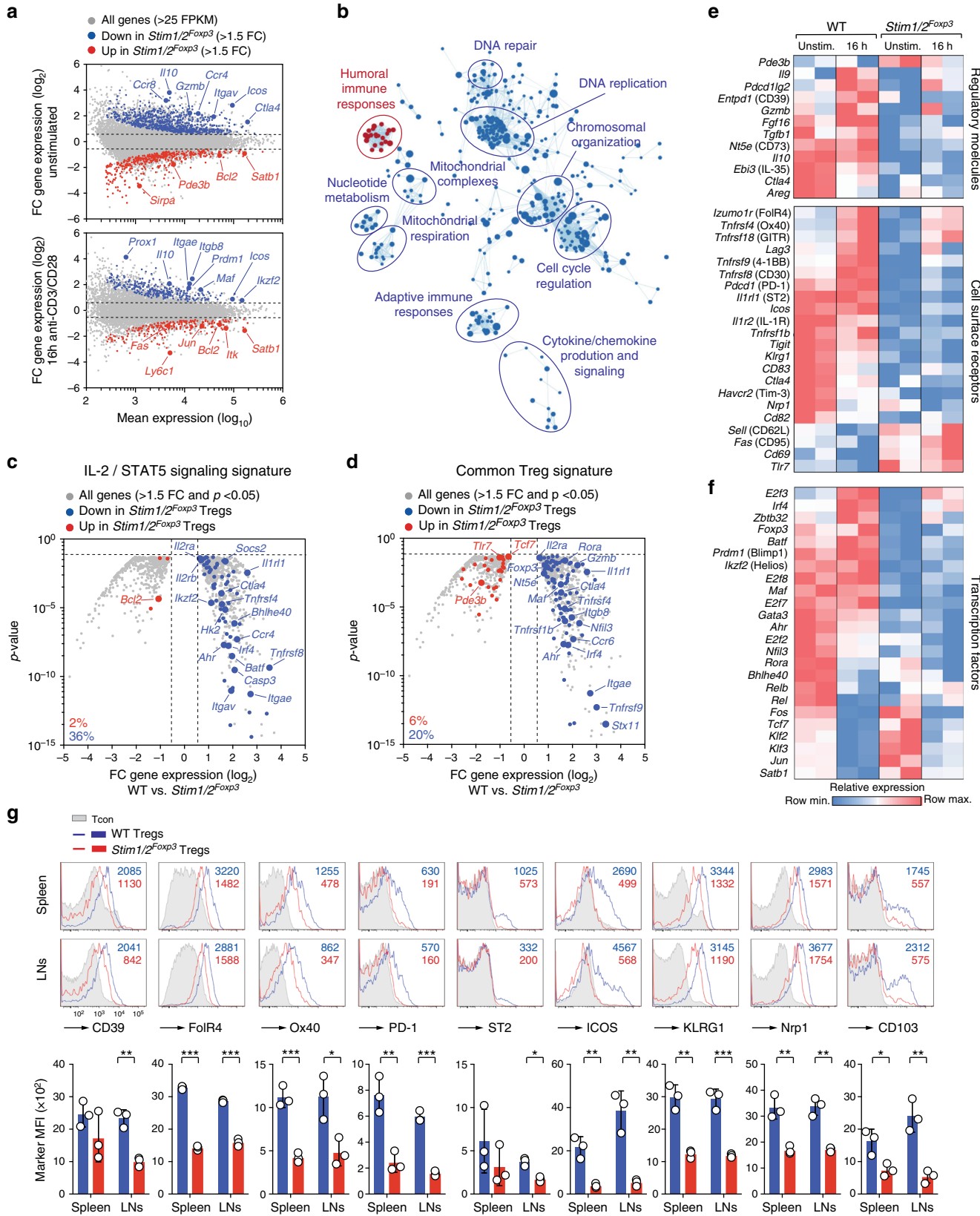

recipient mice (Supplementary Figure 5c) demonstrating that anti-RBC autoantibodies in *Stim1/2^Foxp3* mice cause anemia. To further evaluate the role of self-reactive B cells and autoantibodies in the immunopathology of *Stim1/2^Foxp3* mice, we depleted their peripheral B cells by injection of anti-CD20 antibody into 2–3 weeks old WT and *Stim1/2^Foxp3* mice (Fig. 4j–l). A single injection was sufficient to completely eliminate B cells in the blood of WT and *Stim1/2^Foxp3* mice as early as 4 days post-injection and for more than 4 weeks (Fig. 4j). Ablation of B cells significantly increased RBC numbers and hemoglobin levels in

**Fig. 3** STIM1 and STIM2 control a complex transcriptional network in Treg cells. **a–f** Gene expression analysis of WT and *Stim1/2*-deficient Treg cells isolated from female heterozygous *Stim1/2^Foxp3^* mice by RNA-sequencing. The isolation strategy of Treg cells is shown in Supplementary Figure 2a. **a** MA plots of differentially (>1.5-fold) expressed genes (DEG) in unstimulated WT and *Stim1/2*-deficient Treg cells and Treg cells stimulated with anti-CD3/CD28 for 16 h. Genes significantly (*p* < 0.05) upregulated and downregulated are depicted in red and blue, respectively. **b** Pathway and network analysis based on DEG in unstimulated WT and *Stim1/2*-deficient Treg cells. Downregulated and upregulated pathways are shown in blue and red, respectively. **c** Volcano plot of DEG (gray) between unstimulated WT and *Stim1/2*-deficient Treg cells overlaid with upregulated (red) or downregulated (blue) genes of the 'IL-2/STAT5 signaling signature'[35]. **d** Volcano plot of DEG (gray) between unstimulated WT and *Stim1/2*-deficient Treg cells overlaid with upregulated (red) or downregulated (blue) genes of the 'common Treg signature'[4]. **e** Heatmap analysis of selected DEG (>1.5-fold) encoding regulatory molecules (upper panel) and cell surface receptors (lower panel) in WT and *Stim1/2*-deficient Treg cells. **f** Heatmap of selected differentially expressed (>1.5 fold) transcription factors in WT and *Stim1/2*-deficient Treg cells. **g** Analysis of protein expression of DEG on conventional CD4$^+$ T cells (Tcon), WT and *Stim1/2*-deficient Treg cells by flow cytometry. Bar graphs represent the means ± SEM of four mice. Statistical analysis in (**g**) by unpaired Student's *t*-test. **p* < 0.05; ***p* < 0.01; ****p* < 0.001

*Stim1/2^Foxp3^* mice (Fig. 4k) corroborating the role of autoreactive B cells in the development of AIHA in these mice. Although anti-CD20 treatment ameliorated AIHA, B cell depletion only marginally delayed the premature death of *Stim1/2^Foxp3^* mice (Fig. 4l) indicating that autoantibodies only partly explain their immunopathology. Collectively, our findings demonstrate that SOCE controls the differentiation of Treg cells into Tfr cells and is required to prevent humoral autoimmunity including AIHA.

**Differentiation of tissue-resident Treg cells requires SOCE.** Because B cell depletion was not sufficient to prevent the premature death of *Stim1/2^Foxp3^* mice, SOCE in Treg cells must have additional functions besides controlling Tfr cell differentiation and preventing autoantibody production. Our transcriptome analyses had shown that many molecules typically expressed in effector Treg cells (e.g. 4-1BB, PD-1, ICOS, and GITR) and tissue-resident Treg cells (e.g. ST2, KLRG1, Amphiregulin, and Tim-3)[6,41] were significantly downregulated in *Stim1/2*-deficient compared to WT Treg cells (Fig. 3e, f). We therefore evaluated whether inflammation of non-lymphoid organs contributes to the immunopathology of *Stim1/2^Foxp3^* mice. Leukocytic infiltrates and signs of tissue inflammation were found in the skin, liver, lungs, and BM of all *Stim1/2^Foxp3^* mice examined, whereas inflammation of the pancreas, kidneys, small and large intestine was observed in some but not all animals (Fig. 5a, b). To distinguish between intravascular and tissue-resident immune cells in these organs, we injected WT and *Stim1/2^Foxp3^* mice with fluorescently conjugated anti-CD45 antibodies, which label intravascular but not tissue resident immune cells (Supplementary Figure 6a)[42]. We observed a marked accumulation of CD4$^+$ and CD8$^+$ tissue-resident T cells in the lung (Fig. 5c), liver (Fig. 5d), and skin (Supplementary Figure 6b) of *Stim1/2^Foxp3^* compared to WT mice. No significant accumulation of γδ T cells in the skin of *Stim1/2^Foxp3^* mice was observed (Supplementary Figure 6b). In the liver, dense lymphocytic infiltrates were found in the portal tract surrounding the hepatic bile ducts (Fig. 5b), a finding that is consistent with the presence of AMA (Figure Supplementary 5a) and primary biliary cholangitis (PBC)-like disease, a chronic autoimmune disorder in humans. Inflammatory cytokines were markedly increased in the liver of *Stim1/2^Foxp3^* compared to WT mice with a pronounced elevation of IL-1β and the Th2 cytokines IL-4, IL-9, and IL-13 (Fig. 5e). In line with hepatic inflammation, the levels of alanine aminotransferase (ALT), aspartate aminotransferase (AST), alkaline phosphatase (ALP), and bilirubin in the sera of *Stim1/2^Foxp3^* mice were significantly elevated (Fig. 5f).

Multiorgan inflammation in *Stim1/2^Foxp3^* mice could be due to reduced numbers or function of tissue resident Treg cells. We analyzed the numbers of Foxp3$^+$ Treg cells in the parenchyma of inflamed organs of *Stim1/2^Foxp3^* mice and found a striking decrease in Treg cells in the BM, liver, lungs, and skin of male *Stim1/2^Foxp3^* compared to WT mice (Fig. 6a, Supplementary Figure 6c). This lack of tissue resident Treg cells was in contrast to comparable Treg frequencies in secondary lymphoid organs and blood of WT and *Stim1/2^Foxp3^* mice (Fig. 1c, Fig. 6a) suggesting that SOCE controls the differentiation of tissue resident Treg cells, their maintenance or tissue homing. This interpretation is complicated by the fact that tissue inflammation in male *Stim1/2^Foxp3^* mice may have secondary effects on the numbers of tissue resident Treg cells. To circumvent this limitation, we used two strategies to analyze tissue resident Treg cells in a non-inflamed environment. First, we analyzed WT and *Stim1/2*-deficient tissue resident Treg cells in female heterozygous *Stim1/2^Foxp3^* mice (Fig. 6b) that do not develop autoimmunity (Supplementary Figure 3a). Second, we reconstituted lymphopenic *Rag1^−/−^* mice with BM from CD45.1$^+$ WT and CD45.2$^+$ *Stim1/2^Foxp3^* mice to generate mixed BM chimeras (Fig. 6c). In both models, *Stim1/2*-deficient Treg cells were present in the thymus, spleen, and lymph nodes but were nearly absent in the parenchyma of BM, liver, and lungs (Fig. 6b, d). These results confirm the findings in male *Stim1/2^Foxp3^* mice (Fig. 6a) and demonstrate that SOCE controls tissue residency of Treg cells in a cell-intrinsic manner.

To understand at the molecular level how SOCE controls the development of tissue-resident Treg cells, we re-analyzed our transcriptome data and DEGs in WT and *Stim1/2*-deficient Treg cells (Fig. 3). Many genes encoding transcription factors that promote the differentiation and/or function of tissue-resident Treg cells were markedly downregulated in *Stim1/2*-deficient Treg cells, including *Prdm1* (Blimp-1), *Ahr*, *Rora*, *Rel*, *Irf4*, *Batf*, *Maf*, and *Gata3*[6,12,43,44], whereas regulators that antagonize the formation of tissue-resident lymphocytes and facilitate their tissue egress, such as *Klf2* and *Tcf7* (TCF-1)[43] were upregulated (Fig. 3f). The expression of cell surface receptors that are characteristic of tissue-resident lymphocytes, such as *Il1rl1* (ST2), *Icos*, *Klrg1*, *Havcr2* (Tim-3), and *Tigit*, was impaired in *Stim1/2*-deficient Treg cells (Fig. 3e). In addition, gene expression signatures that are functionally related to tissue homing and trafficking of Treg cells were significantly downregulated in *Stim1/2*-deficient Treg cells (Fig. 3b, Fig. 6e, f). This was mostly due to an impaired expression of chemokine receptors and cell adhesion molecules that control the trafficking of Treg cells into different tissues, including *Itgav* (integrin α$_V$), *Itgae* (integrin α$_E$, CD103), *Itgb8* (integrin β8), *Ccr4*, *Ccr6*, *Cxcr3*, and *Cxcr5* (Fig. 6g, Supplementary Figure 6d)[6]. Taken together, these data suggest that SOCE controls the differentiation of Treg cells into tissue-resident Treg cells and their homing into non-lymphoid organs by orchestrating a network of transcription factors, cell surface, and homing receptors (Supplementary Figure 6e).

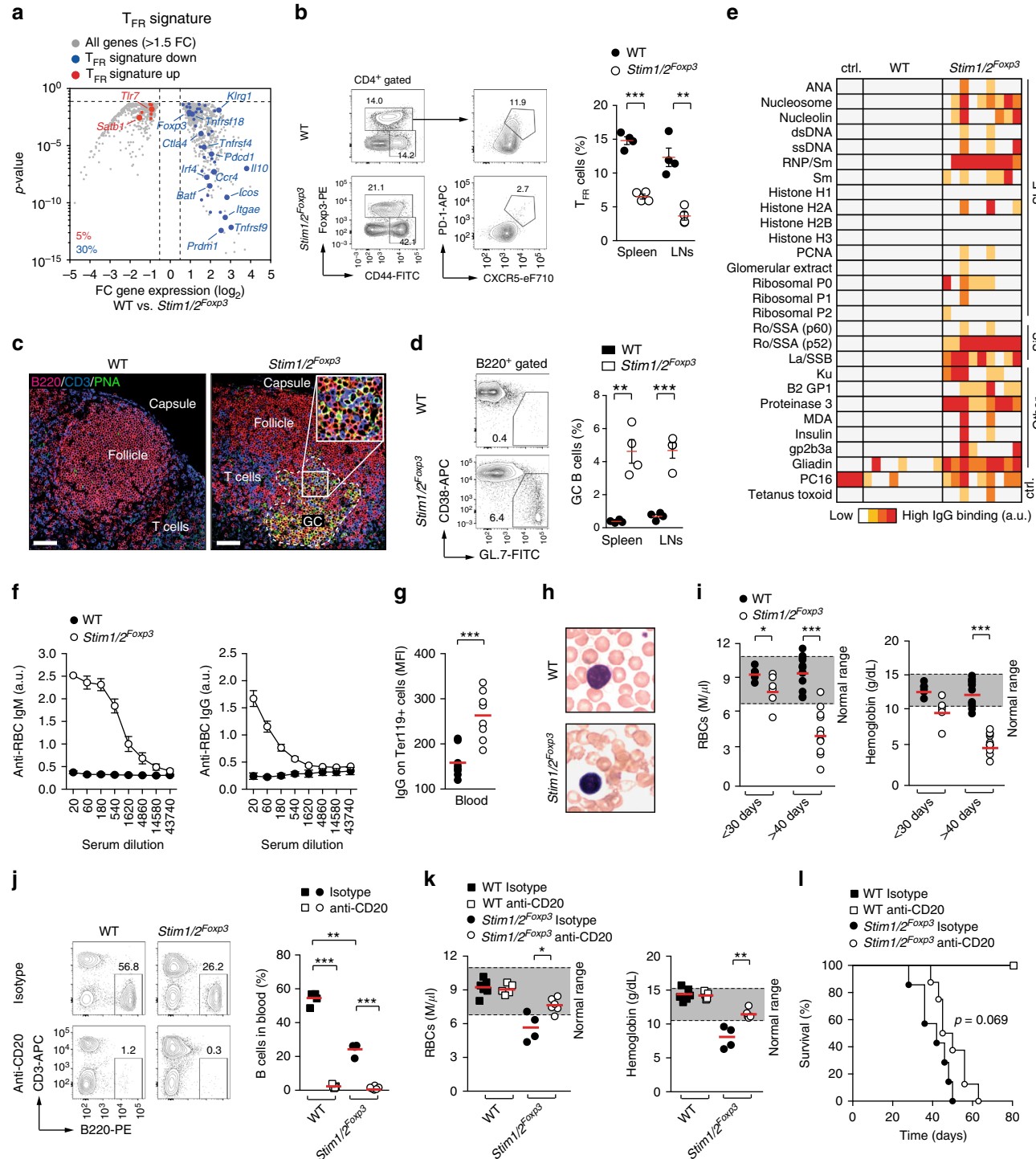

## Discussion

Treg cells are characterized by the expression of Foxp3 that is required for their immunosuppressive function[2,3]. Foxp3 alone is not sufficient, however, for the differentiation and function of Treg cells because ectopic Foxp3 expression in naive CD4[+] T cells does not fully recapitulate the transcriptional signature and function of Treg cells[4], and deletion of *Foxp3* in mature Treg cells does not abrogate key characteristics of Treg cells[5]. These data suggested that additional signaling pathways are required to maintain the unique phenotype and function of Treg cells.

We here show that Ca[2+] influx through CRAC channels initiated by STIM1 and STIM2 activation, in addition to Foxp3, is

critical for the function, lineage identity, and differentiation of mature Treg cells into effector Treg subsets, such as Tfr and tissue-resident Treg cells. CRAC channels control a complex transcriptional network in Treg cells that maintains expression of Treg signature genes and thus their effector functions. This conclusion is in line with the finding that continuous TCR signaling, which presumably includes the activation of CRAC channels, in mature Treg cells is necessary for the sustained expression of numerous Treg signature genes including *Nrp1*, *Izumo1r* (Folr4), *Il2ra*, *Pdcd1* (PD-1), *Ctla4*, *Irf4*, *Ikzf2* (Helios), *Maf*, *Il10*, and *Ebi* (IL-35)[33,34]. We found that expression of the majority of these TCR-induced genes was also dependent on

**Fig. 4** Follicular Treg (Tfr) cell differentiation and prevention of autoantibody production depend on STIM1/STIM2. **a** Volcano plot of DEG in unstimulated WT and *Stim1/2*-deficient Treg cells overlaid with upregulated (red) or downregulated (blue) genes of the 'Tfr cell signature'[8]. **b** Analysis of Tfr cells in the spleen and LNs of male WT and *Stim1/2^Foxp3^* mice; means ± SEM of four mice. **c** Representative immunofluorescence of PNA⁺ germinal centers (GCs) in submandibular LNs of untreated male WT and *Stim1/2^Foxp3^* mice; scale bar: 200 μm. **d** Analysis of GC B cells in male WT and *Stim1/2^Foxp3^* mice; means ± SEM of four mice. **e** Detection of IgG autoantibodies in the sera of male WT and *Stim1/2^Foxp3^* mice using a multiplexed auto-antigen array. Sera of 9–11 individual mice were tested. **f** Analysis of anti-red blood cell (RBC) IgG and IgM autoantibodies using ELISA; means ± SEM of six mice. **g** Detection of relative IgG binding to RBCs of male WT and *Stim1/2^Foxp3^* mice by flow cytometry; means of eight mice. **h** Representative blood smears of WT and *Stim1/2^Foxp3^* mice; scale bar: 10 μm. **i** Analyses of RBC counts and hemoglobin levels in the blood of young (<30 days) and older (>40 days) male WT and *Stim1/2^Foxp3^* mice; means of 6–13 mice. **j–l** Depletion of B cells in *Stim1/2^Foxp3^* mice inhibits RBC destruction but not the premature death of mice. **j** Analysis of blood B cells of male WT and *Stim1/2^Foxp3^* mice 4 days after injection of 1 mg isotype or 1 mg anti-CD20 (5D2) antibody; means of 4–6 mice. **k** RBC counts and hemoglobin levels in the blood of WT and *Stim1/2^Foxp3^* mice 2–4 weeks after antibody injection; means of 4–6 mice. **l** Cumulative survival of WT and *Stim1/2^Foxp3^* mice after antibody injection; 8–10 mice per cohort. Mean survival times (MST): *Stim1/2^Foxp3^* mice isotype control: 42.0 days; *Stim1/2^Foxp3^* mice anti-CD20: 47.5 days; WT mice: >80 days. Each dot in **g**, **j** and **l** represents one mouse. Statistical analysis in **b**, **d**, **g**, **i–k** by unpaired Student's *t*-test; in (**l**) using the Mantel–Cox test. *p < 0.05; **p < 0.01; ***p < 0.001

STIM1/STIM2 suggesting that TCR-mediated Ca²⁺ influx is a major signaling pathway to maintain the transcriptional identity of mature Treg cells.

Our pathway enrichment and network analyses suggest that CRAC channels regulate IL-2 signaling and mTOR-dependent metabolic programming of Treg cells. This is intriguing as Treg cells differ markedly from conventional T cells given their strong reliance on IL-2 and their distinct metabolic properties[36,37,39]. Not only the expression of the IL-2 receptor itself, i.e. *Il2ra* (CD25) and *Il2rb* (CD122), was dependent on STIM1/STIM2 in Treg cells but also an array of genes that are regulated by cytokine signaling, such as *Il1rl1* (ST2), *Ikzf2*, *Irf4*, *Batf*, *Tnfrsf4* (Ox40), and *Ctla4*. The fact that many IL-2-signaling-related genes are dependent on CRAC channels and are part of the common Treg signature suggests that Ca²⁺ influx maintains the lineage identity of Treg cells, at least in part, by modulating their ability to sense IL-2. It is noteworthy that the role of CRAC channels in IL-2 signaling is not limited to maintaining Treg lineage identity, but also involves the development of Treg cells in the thymus. Deletion of *Stim1* and *Stim2* in double-positive thymocytes (by *Cd4*-Cre) or hematopoietic precursor cells (by *Vav*-iCre) significantly impaired the frequencies of Treg cells[21,22]. How STIM1 and STIM2 control the development of Treg cells at the molecular level is not completely understood but given the critical role of IL-2 for Foxp3 induction in Treg precursor cells[45] it is likely that SOCE promotes maturation of tTreg cells indirectly through IL-2 signaling rather than NFAT-mediated Foxp3 expression. This notion is supported by a study reporting decreased IL-2 receptor expression (i.e. CD25 and CD122) in *Stim1/2*-deficient Treg precursor cells and restored induction of Foxp3 expression by administration of high-dose IL-2[21]. It is important to note in this context that while SOCE is essential for Treg development in the thymus and the transcriptional identity of Treg cells, it is not required for the maintenance of Treg cell numbers because deletion of STIM1/STIM2 in post-thymic, mature Treg cells did not perturb the frequencies of Treg cells in secondary lymphoid organs of *Stim1/2^Foxp3^* mice. Note that the increased frequencies of Treg cells we observed in the thymus of *Stim1/2^Foxp3^* mice are likely due to the reimmigration of peripheral Treg cells in response to thymic inflammation as part of the multiorgan inflammation of *Stim1/2^Foxp3^* mice.

We found that expression of many molecules controlling effector functions of Treg cells is dependent on CRAC channels. *Stim1/2*-deficient Treg cells failed to upregulate IRF4, BATF, Blimp-1, and GATA3 that are typically expressed in activated effector Treg cells[46]. In conventional T cells, IRF4, BATF, and GATA3 are crucial for the differentiation of Th2 cells and the production of IL-4. By contrast, Treg cells utilize IRF4, BATF, and GATA3 to suppress Th2 immune responses[47–49]. Abolished

GATA3, IRF4, and BATF expression in *Stim1/2*-deficient Tregs may thus explain the type 2 autoimmunity in *Stim1/2^Foxp3^* mice, which is characterized by elevated Th2 cytokines, increased IgE levels and eosinophilia. Other transcriptional regulators that promote effector functions of Treg cells were also markedly downregulated in the absence of STIM1/STIM2 including the transcription factors Blimp-1, Ahr, c-Maf, Rel B, c-Rel, Bhlhe40, and RORα. Likewise, many surface receptors that are characteristic of activated Treg subsets, such as KLRG1, ST2, OX40, GITR, Lag3, 4-1BB, PD-1, ICOS, Nrp-1, and Tim-3 were reduced in STIM1/2-deficient Treg cells. Together, these data demonstrate that SOCE is important for the differentiation and function of effector Treg cells.

Tfr cells are a specific subset of effector Treg cells and arise from tTreg cells in secondary lymphoid organs[8,9]. *Stim1/2^Foxp3^* mice almost completely lacked Tfr cells resulting in spontaneous GC formation in LNs and spleen, the maturation of self-reactive B cells and production of numerous autoantibodies. The latter included autoantibodies against RBCs causing AIHA in *Stim1/2^Foxp3^* mice. This observation was intriguing because patients with loss-of-function mutations in *STIM1* and *ORAI1* frequently develop humoral autoimmunity including AIHA[26]. Inherited defects in SOCE are rare (~20–30 reported cases worldwide), but several patients from whom blood samples were available showed reduced numbers of circulating Treg cells[26–29,50]. Although Tfr cells could not be evaluated in SOCE-deficient patients, the presence of autoantibodies and AIHA may indicate a Tfr cell defect in these patients. In *Stim1/2^Foxp3^* mice, the lack of Tfr cells is an important contributor to their autoimmunity as depletion of B cells markedly reduced their autoantibody titers and ameliorated AIHA. B cell depletion was not sufficient, however, to prevent the premature death of *Stim1/2^Foxp3^* mice and their severe multiorgan inflammation, suggesting that SOCE regulates other Treg functions besides their differentiation into Tfr cells.

Tissue-resident Treg cells have recently gained much attention because of their pivotal role in the homeostasis of non-lymphoid organs including small and large intestine, skin, lung, liver, adipose tissue, and skeletal muscle. Like Tfr cells, tissue-resident Treg cells differ significantly from their counterparts in lymphoid tissues, for instance by their memory-like phenotype and specific transcriptional signature. The signals that control the tissue residency and organ-specific functions of Treg cells, however, are incompletely understood. *Stim1/2^Foxp3^* mice lacked tissue-resident Treg cells in the lung, liver, BM, and other organs suggesting that SOCE is a critical regulator of the differentiation or recruitment of tissue-resident Treg cells. *Stim1/2*-deficient Treg cells showed reduced expression of many effector molecules and transcription factors normally present in tissue-resident Treg cells

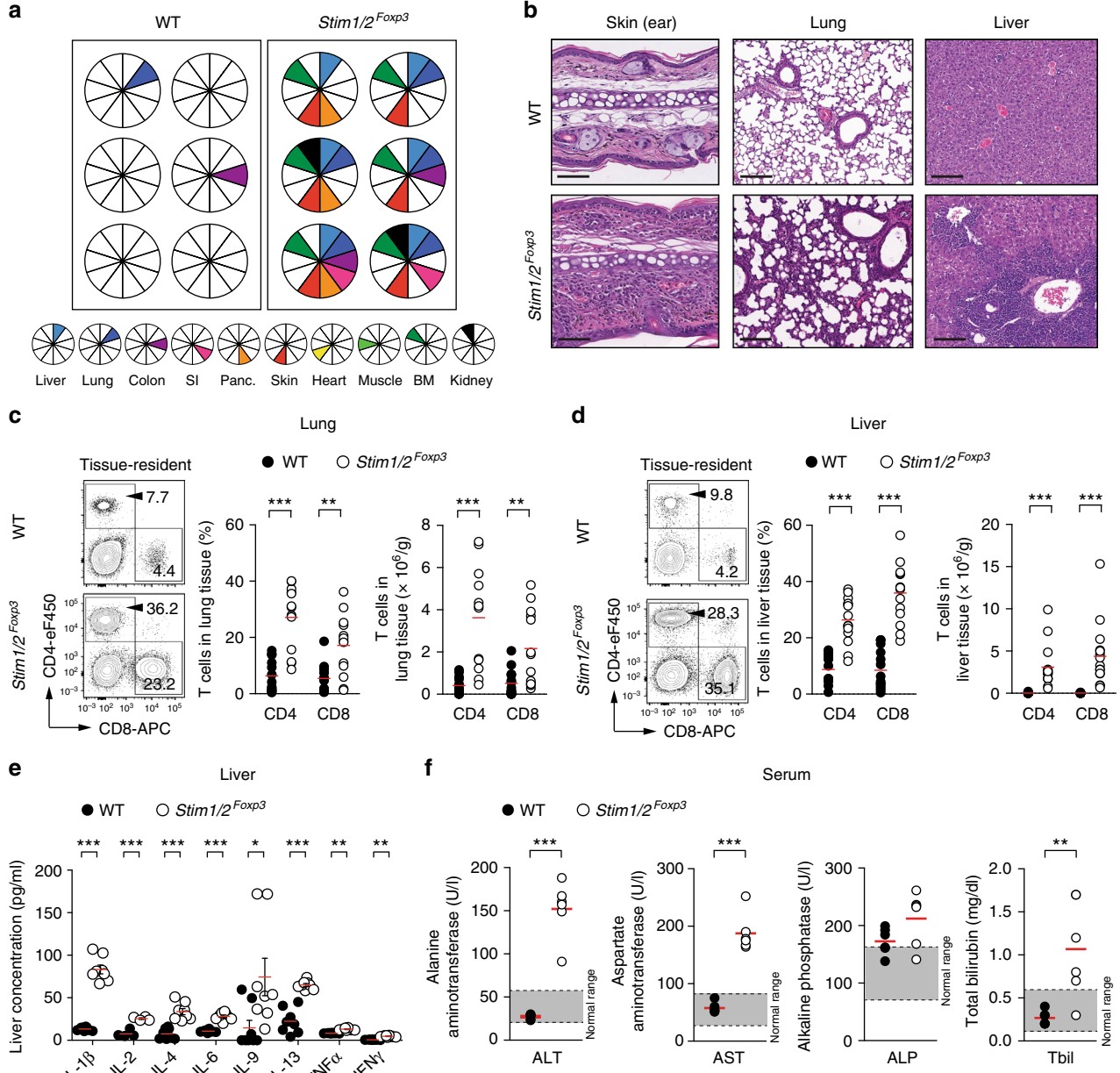

**Fig. 5** Deletion of *Stim1* and *Stim2* genes in Treg cells causes severe tissue inflammation. **a, b** Leukocytic infiltration and signs of tissue inflammation in various non-lymphoid organs of male WT and *Stim1/2*^Foxp3^ mice by histopathology. **a** Each pie chart represents the analysis of one mouse. **b** Representative examples of H&E-stained tissue sections of skin, lung, and liver; scale bars represent 150 μm. **c, d** Analysis of tissue-resident CD4$^+$ and CD8$^+$ T cells in the lung (**c**) and liver (**d**) of male WT and *Stim1/2*^Foxp3^ mice by flow cytometry; means of 12–15 mice. See Supplementary Figure 7 h, i for the gating strategy of tissue-resident T cells. **e** Detection of cytokines in total liver homogenates of male WT and *Stim1/2*^Foxp3^ mice; means ± SEM of 4–8 mice. **f** Analysis of alanine aminotransferase (ALT), aspartate aminotransferase (AST), alkaline phosphatase (ALP), and total bilirubin (Tbil) concentrations in the sera of male WT and *Stim1/2*^Foxp3^ mice by clinical chemistry; means of six mice. Each dot in **c, d,** and **f** represents one mouse. Statistical analysis in **c–f** by unpaired Student's *t*-test. *$p < 0.05$; **$p < 0.01$; ***$p < 0.001$

in skin, skeletal muscle, and VAT[12,13,44]. These molecules included KLRG1, ICOS, IRF4, BATF, Blimp-1 GATA3, IL-10, IL-35, as well as RORα, ST2, and amphiregulin suggesting that SOCE regulates the differentiation of tissue-resident Treg cells. In addition, SOCE controls the expression of chemokine receptors and cell adhesion molecules, such as *Itgav* (integrin α$_V$), *Itgae* (integrin α$_E$, CD103), *Itgb8* (integrin β8), *Ccr4*, *Ccr6*, and *Cxcr3* that may be required for Treg cells to be recruited to or retained in non-lymphoid tissues.

We show here that SOCE controls multiple metabolic pathways in Treg cells implicating SOCE in the metabolic regulation

of Treg function and the differentiation of Tfr cells and tissue-resident Treg cells. Our transcriptome analysis of *Stim1/2*-deficient Treg cells revealed a marked deregulation of genes controlling mTORC1 signaling and metabolic pathways including glycolysis, oxidative phosphorylation (OXPHOS), and mitochondrial electron transport. mTOR signaling controls the proliferation and function of Treg cells, at least in part, by regulating glycolysis and FAO[38–40]. Impaired mTORC1 signaling in the absence of SOCE is consistent with the decreased proliferation of *Stim1/2*-deficient Treg cells in vivo and our previous observation that STIM1 and STIM2 regulate mTOR activity, glycolysis, and

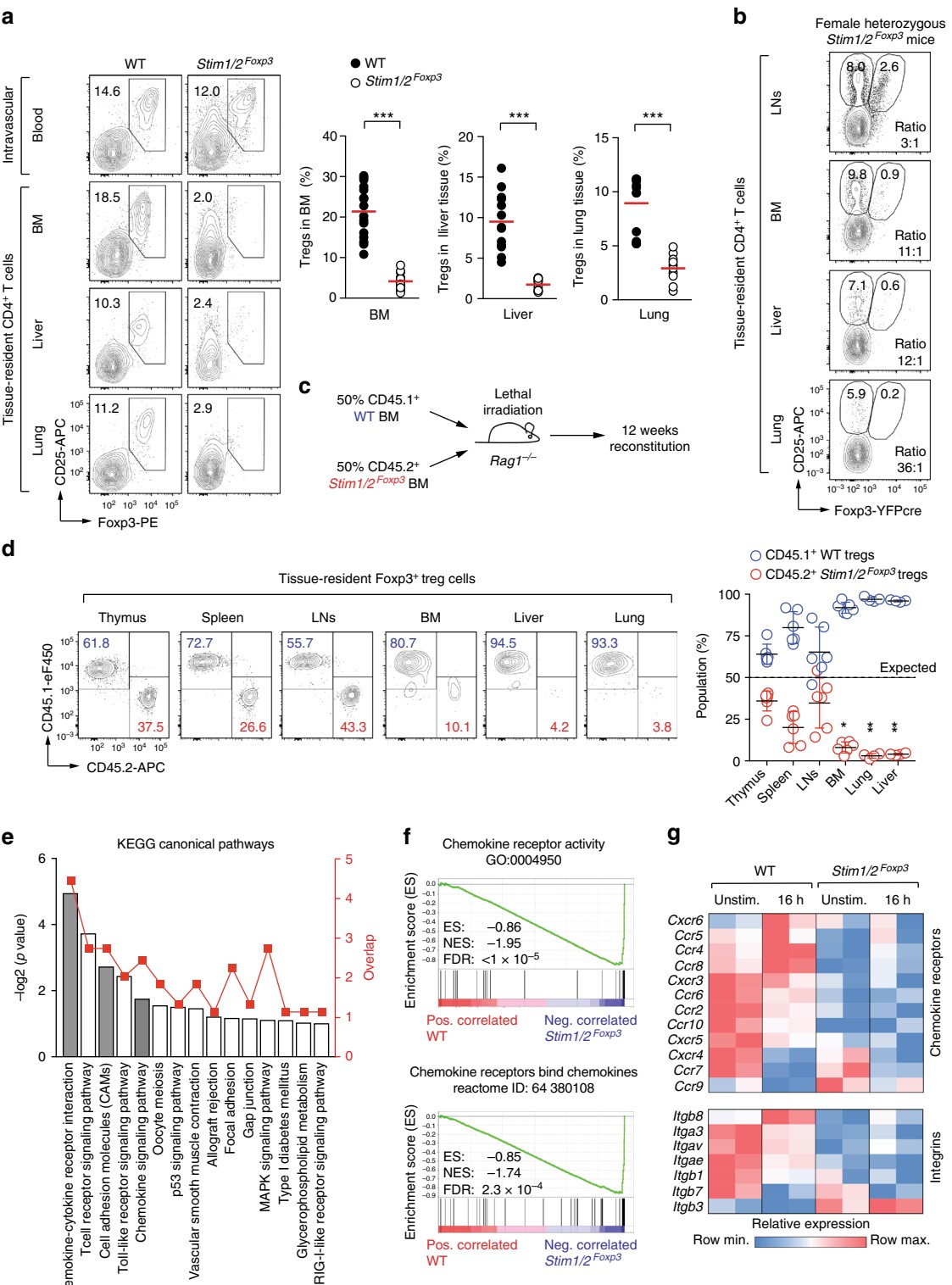

OXPHOS in conventional CD4[+] and CD8[+] T cells[51,52]. It is noteworthy that our analysis of individual genes that are differentially expressed in STIM1/STIM2-deficient compared to WT Treg cells revealed several glycolytic genes (*Aldoc*, *Pkm*, *Pgk1*, and *Hk2*) that we had previously reported to be regulated by SOCE in effector CD4[+] and CD8[+] T cells[52]. However, these represented only a small fraction of all SOCE-regulated genes in effector T cells. These analyses show that while CRAC channels regulate many metabolic genes in Treg cells (this study) and effector

T cells[52], these gene sets are distinct, which may not be surprising given the different metabolic requirements of Treg and Teff cells[53]. Tfr and tissue-resident Treg cells differ significantly from naïve Treg cells in their metabolic properties, and these metabolic adaptations likely contribute to the tissue-specific functions of these Treg subsets. Whereas naive Treg cells have a low glycolytic rate and marked mitochondrial respiration, in vivo studies of effector Treg cells have shown a critical role of mTOR signaling for Treg function through its ability to promote lipid

**Fig. 6** STIM1 and STIM2 control the differentiation and trafficking of tissue-resident Treg cells. **a** Analysis of intravascular and tissue-resident Foxp3+ Treg cells in the parenchyma of BM, liver, and lung of male WT and *Stim1/2*[Foxp3] mice by flow cytometry; means of 10–13 mice. For the gating strategy of tissue-resident Treg cells see Supplementary Figure 7i. **b** Analysis of WT and *Stim1/2*-deficient Treg cells in LNs and the parenchyma of BM, liver, and lung of female heterozygous WT and *Stim1/2*[Foxp3] mice by flow cytometry. Data is representative of three mice; for the gating strategy of tissue-resident Treg cells see Supplementary Figure 7i. **c** Generation of mixed BM chimeric mice using T cell-depleted BM from male CD45.1+ WT mice and BM from male CD45.2+ *Stim1/2*[Foxp3] mice at a 1:1 ratio. **d** Analysis of tissue-resident Treg cells in mixed BM chimeras. Distribution of CD45.1+ WT and CD45.2+ *Stim1/2*-deficient Treg cells in the thymus, spleen, and LNs, as well as the parenchyma of BM, liver, and lung analyzed by flow cytometry; means ± SEM of 4–6 mice. **e–g** Expression of tissue-homing receptors on Treg cells depends on SOCE. **e** Canonical pathway analysis (KEGG) based on differentially expressed genes (DEG) in anti-CD3/CD28 stimulated WT versus *Stim1/2*-deficient Treg cells. **f** Selected gene set enrichment analyses (GSEA) of DEG in WT versus *Stim1/2*-deficient Treg cells. **g** Heatmap analysis of DEG (>1.5 fold) encoding chemokine receptors (upper panel) and integrins (lower panel) in WT and *Stim1/2*-deficient Treg cells. Statistical analysis in **a** and **d** by unpaired Student's *t*-test. *$p < 0.05$; **$p < 0.01$; ***$p < 0.001$

metabolism[39]. The mTOR complex links TCR and IL-2 signals metabolically to Treg activation, and drives the expression of effector molecules that are required for the differentiation of Treg cells into Tfr and tissue-resident Treg cells[39,54,55]. Our finding that SOCE regulates mTORC signaling in Treg cells suggests that Ca$^{2+}$ signals control the differentiation of Tfr cells and tissue-resident Treg cells partly through the metabolic reprogramming of naïve Treg cells.

The transcription factors linking SOCE to the function and differentiation of effector Treg cells are not well understood. In conventional T cells, SOCE controls calcineurin-mediated NFAT activation that promotes the metabolic reprogramming of naïve T cells, their proliferation, and cytokine production[20,52]. Treg cells also require calcineurin signaling because deletion of the calcineurin B subunit in mature Treg cells was shown to cause autoimmunity in mice similar to that observed in *Stim1/2*[Foxp3] mice[56]. Calcineurin-deficient Treg cells had moderately decreased suppressive capacity in vitro and failed to differentiate into effector and tissue-resident Treg cells in vivo[56]. These findings are consistent with the reported NFAT-mediated regulation of Treg cells, which includes the formation of cooperative Foxp3:NFAT complexes that induce CD25 and CTLA-4 expression[57], as well as CXCR5 expression and Tfr cell differentiation[10]. Of the four Ca$^{2+}$/calcineurin-regulated NFAT family members, NFATc1, NFATc2, and NFATc3 are expressed in mature Treg cells but the role of the different NFAT homologs in Treg cells remains controversial[10,24,56,58,59]. Combined deletion of two homologs (either NFATc1 and NFATc2 or NFATc2 and NFATc3) had no major impact on Treg function in vitro and in vivo[24,60]. This is in contrast to conventional T cells, in which ablation of a single NFAT family member markedly impaired effector functions[52,61–63] indicating that Treg cells depend less on individual NFAT homologs than conventional T cells[62]. Besides NFAT, SOCE may also regulate the activity of NF-κB in Treg cells[64,65]. Both the canonical and alternative NF-κB-signaling pathways mediated by p65/Rel A and c-Rel or *Nfkb2*/p52 and Rel B subunits, respectively, were reported to regulate the homeostasis and function of Treg cells and Treg-specific deletion of either *Rela* or *Nfkb2* causes systemic autoimmune disease[66,67]. Interestingly, the activation of the canonical NF-κB pathway is regulated by Ca$^{2+}$ signals in T cells[64,65] suggesting that SOCE may act as an upstream regulator of both NFAT and NF-κB during the differentiation of effector Treg cells. Further studies are required to dissect the Ca$^{2+}$-regulated signaling pathways involved in Treg function and differentiation.

Collectively, our data demonstrate that Ca$^{2+}$ influx via CRAC channels regulates a complex transcriptional network that controls the molecular identity, differentiation, and immunosuppressive function of effector Treg cells, in particular Tfr cells and tissue-resident Treg cells. In the absence of SOCE Treg cells fail to prevent humoral autoimmunity and multiorgan autoimmunity.

## Methods

**Mice**. *Stim1*[fl/fl] and *Stim2*[fl/fl] mice have been described before[22]. *Foxp3-YFPcre* (B6.129(Cg)-*Foxp3*[tm4(YFP/icre)Ayr]/J; JAX strain 016959), *Rag1*[−/−] (B6.129S7-*Rag1*[tm1Mom]/J; JAX strain 002216) and congenic CD45.1+ mice (B6.SJL-*Ptprc*[a]*Pepc*[b]/BoyJ; JAX strain 002014) were purchased from The Jackson Laboratory. Additional *Foxp3-YFPcre* mice were kindly provided by Dr. A. Rudensky. Crossing *Stim1*[fl/fl]*Stim2*[fl/fl] mice to *Foxp3-YFPcre* animals generated mice with Treg-specific deletion of *Stim1/2* genes (*Stim1*[fl/fl]*Stim2*[fl/fl] *Foxp3-YFPcre* or *Stim1/2*[Foxp3] mice for short). All animals were maintained on a pure C57BL/6 genetic background and housed under SPF conditions. All experiments were conducted in accordance with the institutional guidelines for animal welfare approved by the Institutional Animal Care and Use Committee (IACUC) at NYU School of Medicine. Mice received standard rodent chow and autoclaved drinking water ad libitum.

**In vitro and in vivo Treg suppression assay**. $8 \times 10^5$ splenocytes from CD45.1+ congenic mice were labeled with 2.5 μM CFSE (Molecular Probes) and cocultured with $1 \times 10^5$–$1 \times 10^4$ CD45.2+CD4+CD25[hi]YFPcre− (WT) or CD45.2+CD4+CD25[hi]YFPcre+ (*Stim1/2*-deficient) FACS-sorted Treg cells from female heterozygous *Stim1/2*[Foxp3] mice. Cultures were stimulated with 1 μg/ml anti-CD3 (BD Pharmingen, clone 145-2C11) and proliferation of CD4+ and CD8+ responder T cells was measured by flow cytometry 3 days later. To test the suppressive capacity of WT and *Stim1/2*-deficient Tregs in vivo, $5 \times 10^5$ sorted CD45.1+ CD4+CD62L+CD25− T cells were adoptively transferred into lymphopenic *Rag1*[−/−] mice by intraperitoneal (i.p.) injection. After 2–4 weeks, $1 \times 10^5$ purified CD45.2+CD4+CD25[hi]YFPcre− (WT) or CD45.2+CD4+CD25[hi]YFPcre+ (*Stim1/2*-deficient) Treg cells from female heterozygous *Stim1/2*[Foxp3] were transferred i.p. into the same *Rag1*[−/−] mice. *Rag1*[−/−] hosts were assessed twice a week for weight loss and signs of distress as described before[17]. After 12 weeks, CD45.1+ CD4+ donor T cells isolated from the spleen and mesenteric LNs of host mice were analyzed for absolute cell numbers and cytokine production (after PMA/ionomycin restimulation). Absolute numbers of donor-derived CD45.1+ CD4+ conventional T cells (T$_{con}$) were normalized for each of the three independent experiments to account for interexperimental variablity of total T cell numbers in mLNs and to be able to compare the three experimental groups (T$_{con}$ cells only, T$_{con}$ plus WT Treg, T$_{con}$ plus *Stim1/2*-deficient Treg). Briefly, for each independent experiment, the numbers of CD45.1+ T$_{con}$ cells for each experimental group were divided by the average number of T$_{con}$ cells in the T$_{con}$ cell-only samples. The calculated ratios for each sample were then multiplied with the combined average number of CD45.1+ T$_{con}$ cells (in the T$_{con}$-only group) from three experiments to obtain a normalized T$_{con}$ number.

**Flow cytometry and cell sorting**. Staining of cell surface or intracellular antigens with fluorescently labeled antibodies was carried out as described before[50]. Samples were acquired on a LSR II flow cytometer (BD Biosciences) and analyzed using the FlowJo software (TreeStar). Purification of various immune cell populations was performed on a SY3200 cell sorter (Sony) using a 70 μm nozzle. A complete list of antibodies and their respective fluorescent conjugates can be found in Supplementary Table 2. The gating strategies for flow cytometric detection of immune cell populations are summarized in Supplementary Figure 7.

**Analysis of tissue-resident lymphocytes**. To label intravascular immune cells in situ, mice were injected retroorbitally with 1 μg anti-CD45 PE-Cy7 (clone 30-F11) together with 125 U Heparin in 100 μl PBS. After 5 min, mice were sacrificed and lymphoid and non-lymphoid organs were extracted. Skin, lung, and liver tissue (ca. 500 mg) was minced in 2 mm pieces and digested in for 20 min at 37 ℃ using 500 μg/ml Liberase TL (lung, skin) or 25 μg/ml Liberase TM (liver, both Roche) in 5 ml FBS-free RPMI medium. Following digestion, lysates were washed twice and the lymphocyte fraction was enriched by 35% percoll gradient centrifugation before analysis by flow cytometry.

**ELISA, autoantibody arrays, and clinical chemistry analyses**. Ig (IgM, IgG, IgE, and IgE) and cytokine concentrations (i.e. IL-1β, IL-2, IL-4, IL-5, IL-6, IL-9, IL-10,

IL-13, IL-17A, TNFα, and IFNγ) in serum samples and liver homogenizates were analyzed using commercial ELISA kits (all eBioscience) and measured on a Flex-Station 3 plate reader (Molecular Devices). Anti-RBC antibodies were analyzed using RBCs extract (WT RBCs lysed in AKC buffer) bound onto high-binding Maxisorb 96-well plates (Corning). After blocking, serial dilutions of mouse sera were incubated, washed, and bound anti-RBC autoantibodies were detected using AP-conjugated goat-anti-mouse IgM and IgG as detection antibodies (Southern Biotech). To determine IgG antibody levels against a variety of auto-antigens, individual purified proteins, nucleic acids, and complex antigens were each conjugated to individual fluorochrome-conjugated beads in multiplex cocktails of different beads (MagPix, Luminex). Experimental and control group sera were tested at 1:100, 1:1000 and 1:10,000 dilutions, adapting previously described methods[68]. For hematological analyses (i.e. RBCs, hemoglobin levels, and hematocrit), fresh 50 µl heparinized blood samples were analyzed on a veterinary Hemavet 950 blood analyzer (Drew scientific). Liver enzymes concentrations in mouse sera (i.e. ALP, ALT, AST, and total bilirubin) were analyzed by a GLP-certified clinical chemistry laboratory (C-Path, http://cpathlab.com).

**Immunofluorescence and histochemistry.** All tissue specimens were fixed in 4% PFA for at least 24 h and embedded in paraffin. Heat-induced antigen retrieval of tissue sections was performed in 10 mM citric acid buffer (pH 6.0) for 20 min using a high-pressure cooker (Deni). Blocking for 1 h with Antibody Diluent (Dako) was followed by overnight incubation at 4 °C with the primary antibodies rat anti-B220 (1:400, clone RA3-8B2, eBioscience), polyclonal rabbit anti-CD3 (1:100, Dako) and FITC-conjugated PNA (1:100, Vector Laboratories). After washing, anti-CD3 was amplified using biotinylated goat anti-rabbit IgG (1:500, Thermo Scientific) for 40 min at RT. Detection was carried out using donkey anti-rabbit IgG (H+L) AlexaFluor647 (1:800) and streptavidin AlexaFluor488 (1:800) and the Prolong Gold Antifade Mounting Medium containing DAPI (all Molecular Probes). Images were acquired on an automated laser-scanning confocal microscope (Leica TCS SP5 II equipment) and analyzed with the LCS software (Leica) and ImageJ (NIH). Autoantibodies (ANA and AMA) in mouse sera were detected using the KAL-LESTAD HEp-2 cell substrate slides (BioRad). 1:50 dilutions of mouse sera were incubated with HEp-2 slides for 20 min at RT on the slides. After washing, samples were developed using goat anti-mouse IgG (H+L) AlexaFluor488 secondary antibodies (1:800, Molecular Probes) for 20 min. Staining patterns were evaluated on a Nikon Eclipse E600 fluorescence microscope with NIS Elements BR 3.2 (Nikon) software. For hematoxylin and eosin (H&E) staining of non-lymphoid organs, tissue samples were embedded in paraffin blocks, cut into 5 µm sections and stained with H&E as described before[62]. Blood smears were air died for 5 min and fixed immediately with 100% methanol for 5 min before staining with Wright–Giemsa solution (Sigma-Aldrich). H&E and Wright-Giemsa-stained slides were scanned with a SCN400F whole slide scanner and evaluated using the digital image hub software (both Leica).

**Mixed BM chimeras, B cell depletion, and serum transfer.** To generate mixed bone marrow (BM) chimeras, lethally irradiated (11 Gy) $Rag1^{-/-}$ mice were reconstituted with a 1:1 mixture of $2.5 \times 10^6$ WT (CD45.1$^+$) and $2.5 \times 10^6$ $Stim1/2^{Foxp3}$ (CD45.2$^+$) BM and analyzed 10 weeks after reconstitution (see Fig. 6c). B cell depletion in 2–3-week-old WT and $Stim1/2^{Foxp3}$ mice was carried out by i.p. injection of 1 mg anti-CD20 (clone 5D2, Genentech) or isotype control (IgG2a, BioXCell). Successful ablation of B cells was confirmed 4 days later followed by hematological analyses (RBCs, hemoglobin levels, and hematocrit) 3–4 weeks later. To test the presence of anti-RBC autoantibodies, 250 µl pooled serum from WT or $Stim1/2^{Foxp3}$ mice was transferred retro-orbitally into WT recipient mice followed by hematological analyses (RBCs, hemoglobin levels and hematocrit) 3 days later (see Fig. 5b).

**RNA sequencing and data processing.** CD4$^+$CD25$^{hi}$YFPcre$^-$ WT and CD4$^+$CD25$^{hi}$YFPcre$^+$ $Stim1/2$-deficient Treg cells were purified from spleen and LNs of 12–14 pooled female heterozygous $Stim1/2^{Foxp3}$ mice per sample using flow cytometry (for gating strategy see Supplementary Figure 2a). Total RNA was extracted using the RNeasy Micro RNA Isolation Kit (Qiagen). RNA quality and quantity was analyzed on a Bioanalyzer 2100 (Agilent) using a PICO chip. Samples with an RNA integrity number (RIN) of >8 were used for library preparation. RNAseq libraries were prepared using the TruSeq RNA sample prep v2 kit (Illumina), starting from 100 ng of DNAse I (Qiagen) treated total RNA, following the manufacturer's protocol with 15 PCR cycles. The amplified libraries were purified using AMPure beads (Beckman Coulter), quantified by Qubit 2.0 fluorometer (Life Technologies), and visualized in an Agilent Tapestation 2200. The libraries were pooled equimolarly, and loaded on the HiSeq 2500 Sequencing System (Illumina) and run as single 50 nucleotide reads. For gene expression analysis, reads were aligned to the NCBIM37 (iGenome) mouse genome using Bowtie software (Version 1.0.0)[69] with two mismatches allowed. Uniquely mapped reads were further processed by removing PCR duplicates with Picard MarkDuplicates (http://broadinstitute.github.io/picard/). Transcripts were counted using HTSeq and differential gene expression between WT and $Stim1/2$-deficient Treg cells was performed separately for each cell type using DESeq2[70] Bioconductor package in the R statistical programming environment. Differences in gene expression were considered significant if padj < 0.05. Gene expression signatures and canonical pathway analyses and were performed using ingenuity's pathway analysis (IPA, Qiagen), database for annotation, visualization, and integrated discovery (DAVID, https://david.ncifcrf.gov) and gene set enrichment analysis (GSEA, https://software.broadinstitute.org/gsea/index.jsp). For network cluster analyses, positively and negatively regulated gene identifiers were used to generate enrichment maps in Cytoscape (http://www.cytoscape.org). Heatmaps of selected genes were created using the conditional formatting tool in Microsoft Excel. Highest and lowest expression for each gene (row min. and row max.) were displayed as red or blue color, respectively. The RNA-Seq have been deposited in the GEO database under accession number GSE125784.

**Intracellular Ca$^{2+}$ measurement.** Treg cells enriched by flow cytometry were attached to poly-L-Lysine (Sigma Aldrich)-coated translucent 96-well plates (BD Falcon). Cells were loaded with 1 µM Fura-2-AM (Molecular Probes), and washed twice with nominally Ca$^{2+}$-free Ringer solution (containing 155 mM NaCl, 4.5 mM KCl, 3 mM MgCl$_2$, 10 mM D-glucose, 5 mM Na-Hepes (pH 7.4)). Cells were kept in Ca$^{2+}$-free Ringer solution at the beginning of measurements followed by depletion of ER Ca$^{2+}$ stores with 1 µM TG (Calbiochem) after 120 s. At 420 s, Ringer solution containing 2 mM Ca$^{2+}$ was added to cells (1 mM final extracellular Ca$^{2+}$ concentration) to measure SOCE. Fura-2 fluorescence was measured at an emission wavelength of 510 nm after excitation at 340 and 380 nm and plotted as baseline-normalized F340/F380 emission ratio.

**Real-time PCR.** Total RNA was isolated using the RNeasy Micro Kit (Quiagen) and cDNA was synthesized using the iScript cDNA synthesis kit (Bio-Rad). Quantitative realtime (qRT-) PCR was performed using the Maxima SYBR Green qPCR Master Mix (Thermo). Transcripts levels were normalized to the expression of housekeeping genes using the $2^{-\Delta CT}$ method. The complete list of primers used in this study can be found in Supplementary Table 3.

**Statistical analyses.** All results are means with or without the standard error of the means (SEM). The statistical significance of differences between experimental groups was determined by unpaired Student's $t$-test, Mantel-Cox test or one-way ANOVA as indicated in the figure legends. Differences were considered significant for $p$-values < 0.05 (noted in figures as *), $p < 0.01$ (**) and $p < 0.001$ (***). The number of mice per experimental group are indicated in the respective figure legends.

**Reporting summary.** Further information on experimental design is available in the Nature Research Reporting Summary linked to this article.

## Data availability

The authors declare that all data supporting this study can be found within the paper and its Supplementary Information files. The RNA-sequencing dataset is available from the gene expression omnibus (GEO) database at https://www.ncbi.nlm.nih.gov/geo/, accession number GSE125784. Source data for all the charts and graphs can be found in the Source Data file. Additional data supporting the findings of this study are available from the corresponding author (S.F.) upon reasonable request.

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

## Acknowledgements

We thank the Feske lab for many helpful discussions and A. Rudensky for *Foxp3-YFPcre* mice. This work was funded by NIH grants AI097302, AI130143, and AI137004 (S.F.), a pilot grant from the Colton Center for Autoimmunity at NYU School of Medicine

(NYUSOM) (S.F., R.S.L.), P50 AR070591-01A1 and HHSN272201400019C grant (G.J. S.), a NIH/NIDCR grant DE025639 (R.S.L.) and a German Research Foundation (DFG) postdoctoral fellowship VA 882/1-1 (M.V.). The Genome Technology Center, Applied Bioinformatics Laboratories, Histopathology and Cell Sorting core facilities at NYUSOM are partially supported by the Laura and Isaac Perlmutter Cancer Center support grant P30CA016087.

## Author contributions

M.V., Y.-H.W., M.E., J.Y., K.K., R.S.L., G.J.S. and S.F. designed experiments, M.V., Y.-H.W., M.E. and J.Y. conducted experiments, M.V., Y.-H.W., M.E., J.Y., K.K., R.S.L., G.J.S. and S.F. analyzed data and interpreted the results. M.V. and S.F. wrote the manuscript.

## Additional information

**Competing interests:** S.F. is a scientific cofounder of Calcimedica. The remaining authors declare no competing interests.

