## [Peer Review File · Nature Communications]

Reviewers' comments:

Reviewer #1 (Treg, epigenetic regulation, allergy)(Remarks to the Author):

In this study, Vaeth and colleagues provided novel experimental data that demonstrate a role for SOCE in regulating function and differentiation of Treg cells. They initially showed that Treg-specific deletion of STIM1 and STIM2 abolishes SOCE and severely impairs the differentiation and suppressive function of effector Treg cell subsets and causes fatal systemic inflammation. Meanwhile, they demonstrated that SOCE regulates a complex transcriptional network that controls the molecular identity, differentiation and immunosuppressive function of effector Treg cells, in particular Tfr cells and tissue-resident Treg cells that regulate humoral immunity and immune homeostasis in organs and tissues respectively. Transcriptional profiling of STIM1/STIM2-deficient Treg cells revealed that SOCE regulates transcription factors and signaling pathways that control effector Treg differentiation and the maintenance of Treg identity. In the absence of SOCE, Treg cells fail to prevent humoral autoimmunity and autoimmune inflammation of non-lymphoid organs. This is an interesting study and adds to our understanding of the critical role of SOCE in controlling lineage identity and differentiation of effector Treg cells in lymphoid and non-lymphoid tissues. The data are convincing, the conclusions are fair and balanced, and the techniques used in the study are very rigorous. The paper is well written with excellent figures.

The authors have done an excellent job in demonstrating that the Ca²⁺ signals (SOCE) initiated by STIM1 and STIM2, in addition to Foxp3, are critical for the function, lineage identity and the differentiation of mature Treg cells into effector Treg subsets such as Tfr and tissue-resident Treg cells. They also showed that SOCE regulates IL-2 signaling and mTOR-dependent metabolic programming of Treg cells.

I have the following questions/comments:

- The authors mentioned in the paper that while SOCE is essential for the Treg development in the thymus and the maintenance of Treg identity, it is not required for the maintenance of Treg cell numbers as deletion of STIM1 and STIM2 in post-thymic, mature Treg cells did not perturb the frequencies of Treg cells in secondary lymphoid organs of Stim1/2Foxp3 mice. However, in Figure 1C, there's a significant increase of frequency of Treg cells after STIM1 and STIM2 deletions in thymus, lymph nodes and blood. Could the authors explain why?
- There is a typo in the first line of Page 26 "axz nd".
- I also hope the authors (in the future) intend to explore the mechanisms by which STIM1 and STIM2 regulate the development and function of Treg cells at the molecular level, for example, the transcription factors downstream of STIM1 and STIM2 that link SOCE to the function and differentiation of effector Treg cells.

Reviewer #2 (Ca²⁺ signalling, STIM)(Remarks to the Author):

The manuscript by Vaeth et al. describes the effects of a combined deletion of STIM1 and STIM2 in regulatory T cells. While the authors have previously reported on T cell specific ablation of STIM1 and STIM2 which controls thymic development of Treg cells, it remained unclear what the role of store-operated calcium entry was for the differentiation and function of effector Treg cells. Given the known effects of ablation of Tregs on autoimmunity and multi-organ inflammation (i.e. Delgoffe et al, Nature 2013), STIM1/STIM2 deletion caused a similar broad spectrum of autoantibodies and fatal multi-organ

inflammation. Transcriptional profiling and signal pathway analysis reveal a host of SOCE regulated genes and transcription factors that control effector Treg differentiation and the maintenance of Treg identity. Overall, the study is interesting and well performed and contains a host of new pathway analysis data. However, given the previous reports by the authors themselves on SOCE mediated control of metabolic pathways in T cells and several reports on pathway analysis in FoxP3 deprived Tregs, a detailed comparison with these data sets to compare metabolic versus specific Treg functional aspects would have been very interesting.

1. The major concern is that in all figures, Stim1/2Foxp3 derived cells or mice are compared to WT and not to male FoxP3 CRE derived cells or mice. The authors state that in their Stim1/2Foxp3 cells Foxp3 mRNA and protein levels were significantly reduced (Fig. 1A and S2A), although the fraction of cells expressing FoxP3 is partially increased (Fig. 1C) which indicates that insertion of CRE into the Foxp3 locus indeed affects Foxp3 expression and thus may affect Treg function and gene expression, thus some of the smaller effects seen in the Stim1/2Foxp3 cells may not be due to SOCE deletion but rather to a reduced FoxP3 level. To delineate the effects of deletion of SOCE from effects of reduction of FoxP3, the authors can restore expression of FoxP3 by administration of IL2 (as done in Oh-Hora et al., 2013).

2. In Figure S1A and B it is shown that compared to other lymphnodes, the mesenteric LN expand only very little and that the lack of the suppressive effect of Stim1/2Foxp3 Tregs mainly affects Th2 cells. However, in the transplant experiments shown in Figure 2H and 2I, Stim1/2Foxp3 Tregs failed to suppress conventional T cells in mesenteric LNs (so should lead to their expansion, discrepancy to Fig. S1A,B) and their differentiation into Th1 and Th17 cells (text), but IL-17 and IFN γ levels are increased. This is somewhat confusing as Figure 2D shows the major effects on Th2 cytokines. Why did the authors not measure the levels of IL4, IL5 and TNF α in the transplanted mice? And why was not an asthma model where Th2 play a major role used?

3. For a better comparison with the human patients harbouring STIM1 LOF mutations, why did the authors not work with only STIM1 deficient Tregs? Why was the transcriptome analysis not performed with human patient and control Tregs?

4. In Figure S3B, the control of SOCE measurements in the Tcon controls isolated in S3A is missing as was done in Figure 1B. Minor: In both figures the time before readdition of Calcium is too short, as the intracellular stores are not completely depleted.

5. In Figure 1D the statistics are missing, the authors state that the populations are not shifted but the presented numbers show changes ranging from 12 – 25%.

6. On page 5, the authors state that there is not difference in Tbet or ROR γ T expression (data not shown), however, this data is very relevant considering the observed increase in IFN γ and IL17 expression and should therefore be presented.

7. No statistics are given in Figure 3G to show how significant the changes in Treg specific genes are. Minor:

Page 4 bottom typo: by -> but, same sentence contains wrong Figure reference (Figure 1E should be 1F)

In Figure 2C the label colors in CD4⁺/FoxP3⁻ cells are switched (compare numbers with bar graph in right hand panel).

Point by point response

We thank the reviewers for their careful evaluation of the manuscript and helpful comments. Before providing a point-by-point response to the critiques, please note that the order of supplemental Figures S1-S3 has changed because we realized that these figures were referenced out of order in the results text.

Reviewer #1

The authors mentioned in the paper that while SOCE is essential for the Treg development in the thymus and the maintenance of Treg identity, it is not required for the maintenance of Treg cell numbers as deletion of STIM1 and STIM2 in post-thymic, mature Treg cells did not perturb the frequencies of Treg cells in secondary lymphoid organs of Stim1/2Foxp3 mice. However, in Figure 1C, there's a significant increase of frequency of Treg cells after STIM1 and STIM 2 deletions in thymus, lymph nodes and blood. Could the authors explain why? **Response:** The most likely explanation for increased frequencies of Treg cells in lymph nodes and blood (**Figure 1c**) is that STIM1/STIM2-deficient Treg cells cannot migrate into non-lymphoid tissues and may therefore get stuck in the lymphoid organs and blood. This explanation is consistent with a key finding of our paper, which are reduced numbers of tissue-resident Treg cells (Figure 6). Molecules that allow Treg cells to migrate into tissues such as chemokine receptors, and integrins are reduced in the absence of STIM1/STIM2 whereas other molecules that recruit Treg cells to lymphoid organs or retain them are increased (CD62L, CCR7, KLF2, TCF7). In addition, inflammatory cytokines may also drive Treg expansion in secondary lymphoid organs and blood.

Reviewer Figure 1. Thymic inflammation and atrophy in *Stim1/2^{FOXP3}* mice. (A) Flow cytometric analysis of CD4⁺ and CD8⁺ expression of thymocytes from 4-6 weeks old male *Stim1/2^{FOXP3}* mice. **(B)** Quantification of total thymocyte numbers of WT and 4-6 weeks old male *Stim1/2^{FOXP3}* mice. Data represent the mean of 13-14 mice. **(C)** Representative image of the thymus of WT and *Stim1/2^{FOXP3}* mice shows thymic atrophy. For details see text. Statistical analysis by unpaired Student's t test. ***, p<0.001.

A likely explanation for a relative increase in the frequencies of Treg cells in the thymus of *Stim1/2^{FOXP3}* mice (**Figure 1c**) is thymic inflammation (as part of the multiorgan inflammation in *Stim1/2^{FOXP3}* mice) and T cell immigration into the thymus. Inflammation of the thymus is consistent with (i) the thymic atrophy observed in *Stim1/2^{FOXP3}* mice, and (ii) the increased frequencies of single-positive CD4⁺ and CD8⁺ T cells in their thymus, which most likely are peripheral T cells that have re-entered the thymus and not immature SP T cells). As a result of the thymus atrophy, the absolute numbers of Treg cells in the thymus of *Stim1/2^{FOXP3}* mice are reduced compared to WT mice (**Reviewer Figure 1**). Please note that absolute Treg numbers are only reduced in male *Stim1/2^{FOXP3}* mice with multiorgan inflammation, but not in WT: *Stim1/2^{FOXP3}* mixed bone marrow chimeric mice (**Figure 6d**), that lack inflammation and in which Treg development is normal. We have added a sentence discussing these findings to the Discussion section.

• *There is a typo in the first line of Page 26 "axz nd".* **Response:** This mistake has been corrected.

• I also hope the authors (in the future) intend to explore the mechanisms by which STIM1 and STIM2 regulate the development and function of Treg cells at the molecular level, for example, the transcription factors downstream of STIM1 and STIM2 that link SOCE to the function and differentiation of effector Treg cells. **Response:** We agree with the reviewer that this is an important future direction of research. We are indeed planning a comprehensive chromatin-level analysis how Ca²⁺ signals through CRAC channels regulate gene expression in CD4⁺ T cells including Treg cells, which includes an analysis of transcription factors that are activated by Ca²⁺ signals downstream of CRAC channels. Since the reviewer is not requesting such experiments for the revision of this paper, we have not prioritized such experiments, which will take more time than would be allowed for the revision.

Reviewer #2

Given the previous reports by the authors themselves on SOCE mediated control of metabolic pathways in T cells and several reports on pathway analysis in FoxP3 deprived Tregs, a detailed comparison with these data sets to compare metabolic versus specific Treg functional aspects would have been very interesting. **Response:** We had previously shown that SOCE mediated by STIM1/STIM2 is required for the induction of aerobic glycolysis in stimulated T cells by regulating the expression of transcription factors, glucose transporters and glycolytic enzymes (Vaeth et al. *Immunity* 2017, PMID: 29030115). These studies were limited to effector CD4⁺ and CD8⁺ T cells and did not include Treg cells. Treg cells are thought to be less dependent on glycolysis than T_{eff} cells and utilize instead fatty acid oxidation and oxidative phosphorylation (OXPHOS) by mitochondria for their metabolic needs (e.g. Gerriets & Rathmell, 2012, PMID: 22342741). We therefore did not necessarily expect differences in glycolytic and other metabolic pathways between Treg cells from WT and *Stim1/2*^{FOXP3} mice.

In this study, we provide novel evidence that SOCE is also required for mitochondrial metabolism in Treg cells, because genes regulating mitochondrial respiration and mitochondrial complexes were downregulated in Treg cells from *Stim1/2*^{FOXP3} mice (**Figure 3b** and **Figure S4a**). Our gene set enrichment analysis (GSEA) shows that the expression of genes associated with OXPHOS, the mitochondrial respiratory chain and the TCA cycle are impaired in Treg cells from *Stim1/2*^{FOXP3} mice (**Figure S4e**). In addition, and notwithstanding the important role of mitochondrial function in Treg cells, we also found a reduced expression of genes associated with glycolysis (**Figure S4e**) and mTORC1 signaling (**Figure S4b,c**), which is an essential regulator of cell growth linked to aerobic glycolysis.

Reviewer Figure 2. Metabolic gene expression in WT and *Stim1/2*-deficient Treg cells. (a,b) Expression of metabolic genes that we previously identified to be (a) SOCE-dependent in CD4⁺ and CD8⁺ effector T cells (Vaeth et al., *Immunity* 2017, PMID: 29030115) and (b) SOCE-dependent in fibroblasts and other non-immune cell types (Maus et al., *Cell Metabolism* 2017, PMID: 28132808). (c,d) Expression of metabolic genes that are thought to be (c) specific for Treg cells (based on Hill et al., *Immunity* 2007, PMID: 18024188 and presented in Figure 3d) and metabolic genes that are (d) induced by IL-2 / STAT5 signaling in Treg cells (based on the molecular gene expression signature presented in Figure 3c). Genes that are differentially expressed between WT and *Stim1/2*-deficient Treg cells are marked with asterisks, *, p<0.05; ***, p<0.001; n.s., not significant.

In response to the reviewer's comment, we have expanded our analysis and provide additional information regarding the role of SOCE in the expression of metabolism-related genes in Treg cells. We first compared the expression of glycolytic genes that we had found to be SOCE-dependent in effector CD4⁺ and CD8⁺ T cells (Vaeth et al., *Immunity* 2017, PMID: 29030115) to the RNA-Seq results in WT and STIM1/STIM2-deficient Treg cells presented in this study. Only a small fraction of genes that are STIM1/STIM2-dependent in effector T cells were differentially expressed in STIM1/STIM2-deficient Treg cells compared to WT Treg cells (*Aldoc*, *Pkm*, *Pgk1* and *Hk2*) (**Reviewer Figure 2a**). Given the different metabolic requirements of Teff and Treg cells, these findings are not too surprising,

We further compared the expression of SOCE-dependent genes that are involved in mitochondrial and lipid metabolism in non-immune cells based on a recent study from our lab (Maus et al., *Cell Metabolism* 2017, PMID: 28132808). Only the expression of a single gene (*Cox4i1*) was differentially expressed in STIM1/STIM2-deficient vs. WT Treg cells (**Reviewer Figure 2b**). However, not only do the experimental conditions differ between these two studies, SOCE-regulated metabolic genes in non-immune cells may not be relevant for the metabolic function of Treg cells.

Lastly, we analyzed whether Treg-specific and/or Foxp3-dependent metabolic genes are regulated by SOCE. To this end, we analyzed the expression of genes in WT and STIM1/STIM2-deficient Treg cells that (i) were identified to be Treg-specific (based on Hill et al., *Immunity* 2007, PMID: 18024188 and presented in Figure 3d of the manuscript) (**Reviewer Figure 2c**), (ii) dependent on IL-2 / STAT5 signaling in Treg cells (based on the molecular gene expression signature presented in Figure 3c) (**Reviewer Figure 2d**) or (iii) are involved in primary metabolic processes and differentially expressed upon Foxp3 deletion (based on Gerriets et al. *Nat. Immunol.* 2017, PMID: 27695003) (**Reviewer Figure 3**).

Reviewer Figure 3. Expression of Foxp3-dependent metabolic genes in WT and *Stim1/Stim2*-deficient Treg cells. Expression of genes that are involved in primary metabolic processes and differentially expressed upon inducible Foxp3 deletion (based on Gerriets et al. *Nat. Immunol.* 2017, PMID: 27695003). Genes that are differentially expressed between WT and *Stim1/Stim2*-deficient Treg cells (*Acss2*, *Alox8*, *Ahr*) are marked with asterisks. *, p<0.05; **, p<0.005***, p<0.001; n.s., not significant.

The latter results show that only 3 of the Foxp3-dependent metabolic genes are differentially expressed in STIM1/STIM2-deficient Treg cells. They include *Acss2* (Acyl-CoA Synthetase Short Chain Family Member 2, which catalyzes the activation of acetate for use in lipid synthesis and energy generation), *Alox8* (arachidonate 8-lipoxygenase, which catalyzes the addition of molecular oxygen to polyunsaturated fatty acids to yield fatty

acid hydroperoxides) and *Ahr* (Aryl Hydrocarbon Receptor, which is a ligand-activated transcription factor that has been shown to regulate xenobiotic-metabolizing enzymes such as cytochrome P450).

Collectively, these analyses suggests to us that while STIM1/STIM2 and SOCE regulate many metabolic genes in Treg cells (based on our GSEA and pathway analysis shown in **Figure 3** and **Figure S4** of this study), these genes are fundamentally different from the SOCE-regulated metabolic genes in effector T cells (Vaeth et al., *Immunity* 2017, PMID: 29030115) and fibroblasts (Maus et al., *Cell Metabolism* 2017, PMID: 28132808). Given the reported different metabolic requirements of Teff, Treg and non-immune cells, these findings are not too surprising, but emphasize a universally important role of SOCE in the regulation of different metabolic pathways in different cell types. We have added several sentences discussing these findings and conclusions to the Discussion section.

1. The major concern is that in all figures, Stim1/2Foxp3 derived cells or mice are compared to WT and not to male FoxP3 CRE derived cells or mice. The authors state that in their Stim1/2Foxp3 cells Foxp3 mRNA and protein levels were significantly reduced (Fig. 1A and S2A), although the fraction of cells expressing FoxP3 is partially increased (Fig. 1C) which indicates that insertion of CRE into the Foxp3 locus indeed affects Foxp3 expression and thus may affect Treg function and gene expression, thus some of the smaller effects seen in the Stim1/2Foxp3 cells may not be due to SOCE deletion but rather to a reduced FoxP3 level. To delineate the effects of deletion of SOCE from effects of reduction of FoxP3, the authors can restore expression of FoxP3 by administration of IL2 (as done in Oh-Hora et al., 2013). **Response:** We agree with the reviewer that it is important to determine whether reduced Foxp3 levels in Treg cells of *Stim1/2^{FOXP3}* mice are due to abolished SOCE or whether insertion of the IRES-YFP/Cre cassette into the Foxp3 locus affects Foxp3 expression. Adverse effects on Foxp3 expression in Treg cells of mice with targeted insertion of fluorochrome reporter genes into the endogenous Foxp3 gene have been reported (PMID: 22879902). This analysis, however, did not include the Foxp3-YFP/Cre (B6.129(Cg)-Foxp3tm4(YFP/cre)Ayr/J, JAX 016959) mice used in our study. We found evidence in the literature that insertion of Foxp3-YFP/Cre into the Foxp3 locus has no effect on Foxp3 expression (PMID: 26304965, *Figure 4B*, *Figure 7A* and PMID: 21468021, *Figure 6A*), but also evidence to the contrary, i.e. slightly reduced Foxp3 levels in YFP⁺ vs YFP⁻ T cells of Foxp3-YFP/Cre mice (PMID: 29661826, *Figure 3 & 4*).

In response to the reviewer's request, we therefore conducted several experiments.

(1) We compared Foxp3 expression in Treg cells from female C57BL/6 WT mice, female homozygous *Foxp3-YFP/Cre* mice without "floxed" *Stim1* and *Stim2* alleles (JAX stock 016959) and female hemizygous *Stim1^{fl/fl}Stim2^{fl/fl}Foxp3-YFP/Cre^{+/-}* (*Stim1/2^{FOXP3}*). Foxp3 protein levels were analyzed by antibody staining for Foxp3 and YFP and analyzed by flow cytometry. (i) We found that Foxp3 levels are moderately but significantly reduced in YFP⁺ Treg cells of *Foxp3-YFP/Cre* mice compared to YFP⁻ Treg cells of C57BL/6 WT mice (**new Figure S1b**). (ii) Furthermore, we found that Foxp3 levels are even more significantly reduced in YFP⁺ Treg cells of *Stim1^{fl/fl}Stim2^{fl/fl}Foxp3-YFP/Cre* mice compared to their YFP⁻ Treg cells (**new Figure S1b**). When we compared Foxp3 levels in YFP⁺ Treg cells of *Foxp3-YFP/Cre* (SOCE intact) and *Stim1^{fl/fl}Stim2^{fl/fl}Foxp3-YFP/Cre^{+/-}* (no SOCE) mice, we found a significant ($p < 0.01$) reduction of Foxp3 expression in SOCE-deficient compared to SOCE-competent Treg cells (**new Figure S1b**, compare green and red columns). Taken together, these data show that insertion of IRES-YFP/Cre in the Foxp3 locus does have a moderate effect on Foxp3 expression, but that deletion of *Stim1* and *Stim2* genes further reduces Foxp3 levels. SOCE therefore is required for full Foxp3 expression in post-thymic Treg cells.

(2) Furthermore, we analyzed Foxp3 expression in WT and *Stim1^{fl/fl}Stim2^{fl/fl}Cd4Cre* mice. In *Stim1^{fl/fl}Stim2^{fl/fl}Cd4Cre* mice, the Cre transgene is located on a different chromosome as the *Foxp3* gene locus can therefore not interfere directly with Foxp3 expression. Similar as in *Stim1^{fl/fl}Stim2^{fl/fl}Foxp3-YFP/Cre* mice, we also found a reduced Foxp3 protein levels in Treg cells from *Stim1^{fl/fl}Stim2^{fl/fl}Cd4Cre⁺* compared to *Cd4Cre⁻* WT control mice (**Reviewer Figure 4**). These findings are furthermore consistent with the previously reported reduction in Foxp3 levels in Treg cells from *Stim1^{fl/fl}Stim2^{fl/fl}Vav-iCre* mice (PMID: 23499491, *Figure 5B*), in which the Vav-iCre transgene is integrated on chromosome 18 (not the Foxp3 locus on the X chromosome) [<https://www.jax.org/strain/008610>] resulting in deletion of STIM1/STIM2 in all immune cells. Collectively, these findings support the conclusion that the deletion of *Stim1* and *Stim2*, and thus SOCE, is directly responsible for reduced Foxp3 levels.

Reviewer Figure 4. Reduced Foxp3 protein expression in Treg cells of *Stim1^{fl/fl}Stim2^{fl/fl} Cd4Cre* mice. Representative histogram plots from flow cytometry experiments measuring Foxp3 expression in Treg cell as geometric mean fluorescence intensity (gMFI) in WT and *Stim1^{fl/fl}Stim2^{fl/fl} Cd4Cre* mice (left panels). Means \pm SEM of 5-7 mice per group (right panel). * $p < 0.05$. Statistical analysis by unpaired Student's t test.

(3) Following the reviewers suggestion "the authors can restore expression of FoxP3 by administration of IL2 (as done in Oh-Hora et al., 2013)", we tested the effects of IL-2 on Foxp3 levels in Treg cells. We stimulated Treg cells isolated from C57BL/6 WT mice, female homozygous *Foxp3-YFP/Cre* mice without "floxed" *Stim1* and *Stim2* alleles (JAX stock 016959) and female hemizygous *Stim1^{fl/fl}Stim2^{fl/fl} Foxp3-YFP/Cre^{+/-}* (*Stim1/2^{FOXP3}*) in the absence or presence of 500 U/ml IL-2 for 24 h. We found that the reduced Foxp3 levels in Tregs of *Foxp3-YFP/Cre* compared to WT mice were rescued by IL-2 (**Reviewer Figure 5**). By contrast, IL-2 did not rescue Foxp3 expression in YFP⁺ Treg cells of *Stim1^{fl/fl}Stim2^{fl/fl} Foxp3-YFP/Cre^{+/-}* mice (no SOCE) compared to YFP⁻ Treg cells of the same mice (intact SOCE) (**Reviewer Figure 5**). The failure of IL-2 to rescue Foxp3 expression is consistent with our finding that the 'IL-2 / STAT5 signaling pathway' is downregulated in the absence of *Stim1/2* in Treg cells (**Figure 3c**) resulting in a defect in sensing IL-2 and a failure to respond to exogenous IL-2. This defect in IL-2 sensing may in fact explain the reduced Foxp3 expression levels in STIM1/STIM2-deficient Treg cells (in either *Foxp3-YFP/Cre*, *CD4-Cre* or *Vav-iCre* as discussed above) and it is consistent with the role of IL-2 in maintaining Foxp3 expression shown before (e.g. in Chinen et al. *Nat. Immunol.* 2016 PMID: 27595233, Fontenot *Nat. Immunol.* 2005 PMID: 16227984, and more recently in Fan et al., *Cell Reports* 2018 PMID: 30380412).

Reviewer Figure 5. STIM1/STIM2-deficient Treg cells fail to upregulate Foxp3 expression in response to exogenous IL-2. Foxp3 expression levels in WT (white and blue bars), female homozygous *Foxp3-YFP/Cre* mice (SOCE intact; green bars) and female hemizygous *Stim1^{fl/fl}Stim2^{fl/fl} Foxp3-YFP/Cre^{+/-}* (*Stim1/2^{FOXP3}*; SOCE deleted, red bars) in the absence (left panel) or presence of 500 U/ml IL-2 for 24h (right panel). Foxp3 expression was analyzed by flow cytometry. Shown are means \pm SD of 3 mice per group. * $p < 0.05$; ** $p < 0.01$; *** $p < 0.001$. Statistical analysis by unpaired Student's t test. We conclude that whereas IL-2 can elevate Foxp3 expression in Treg cells from *Foxp3-YFP/Cre* mice (SOCE intact) no response to IL-2 was observed in Treg cells from *Stim1^{fl/fl}Stim2^{fl/fl} Foxp3-YFP/Cre^{+/-}* mice.

Taken together, we conclude that defective Treg differentiation and function in *Stim1/2 Foxp3-YFPCre* (*Stim1/2^{FOXP3}*) mice is mainly due to abolished SOCE and not insertion of IRES-YFP/Cre in the Foxp3 locus. Although abolished SOCE does have a small effect on Foxp3 expression levels in Treg cells, it is important to note that lack of *Stim1/2* and SOCE has a more profound effect on gene expression programs that regulate Treg differentiation and function (**Figure 3**).

2. In Figure S1A and B it is shown that compared to other lymphnodes, the mesenteric LN expand only very little and that the lack of the suppressive effect of *Stim1/2Foxp3* Tregs mainly affects Th2 cells. However, in the transplant experiments shown in Figure 2H and 2I, *Stim1/2Foxp3* Tregs failed to suppress conventional T cells in mesenteric LNs (so should lead to their expansion, **discrepancy** to Fig. S1A,B) and their differentiation into Th1 and Th17 cells (text), but IL-17 and IFN γ levels are increased. This is somewhat confusing as Figure 2D shows the major effects on Th2 cytokines. Why did the authors not measure the levels of IL4, IL5 and TNF α in the transplanted mice? And why was not an asthma model where Th2 play a major role used? **Response:** Regarding the only slightly enlarged mesenteric LNs in *Stim1/2^{FOXP3}* mice but increased numbers of conventional T cells in mesenteric LNs after adoptive co-transfer with *Stim1/2*-deficient Treg cells (**Figure 2h**), we would like to point out that the image shown in **Figure S1c** represents only one mouse, and that we have indeed observed mice with enlarged mesenteric LNs. More importantly, we want to emphasize that the effects of *Stim1/2*-deficient Treg cells on mesenteric LNs of *Stim1/2^{FOXP3}* mice (**Figure S1c**) occur in a very different context than in lymphopenic *Rag1^{-/-}* mice after adoptive transfer of Treg cells plus effector T cells (**Figure 2h**). The atrophic mesenteric LNs of host *Rag1^{-/-}* mice are populated by proliferating transferred donor T cells, whose expansion is suppressed by WT but not *Stim1/2*-deficient Treg cells (**Figure 2h**). By contrast, the mesenteric LNs of *Stim1/2^{FOXP3}* mice are not "empty" unlike those of *Rag1^{-/-}* mice and homeostatic proliferation of effector T cells does not occur (or to a lesser degree) compared to the LNs of *Rag1^{-/-}* mice. Collectively, we therefore do not think that there is a discrepancy between data shown in Figure S1a and Figure 2h.

Regarding the predominance of increased Th2 cytokine levels in Figure 2d and IFN γ /IL-17A levels in Figure 2h-i, we would like to point out that cytokines were measured under different conditions. In Figure 2d, we measure spontaneous cytokine levels in the serum of male *Stim1/2^{FOXP3}* mice. In this scenario cytokines not only originate from effector T cells but also from other immune cell types that are normally suppressed by functional Treg cells and can produce type 2 cytokines (e.g. mast cells, basophils). In Figure 2h-i, we adoptively transfer T cells to *Rag1^{-/-}* mice, which favors the differentiation of Th1 and Th17 cells in the gut. It would be unusual to find Th2 cytokines produced by adoptively transferred T cells that migrate to the gut mucosa. We therefore do not think that the data shown in Figure 2d and 2h-i are inconsistent. They merely reflect different types of T cell responses in different tissues and experimental models that are both regulated by Treg cells.

Regarding the suggestion to study *Stim1/2^{FOXP3}* mice in an asthma model, we agree that this would be interesting to test. *Stim1/2^{FOXP3}* mice spontaneously develop lung inflammation with features similar to those found in asthma including immune cell infiltration into the lung parenchyma, eosinophilia and high IgE titers at 4-6 weeks of age. To induce asthma by immunization with ovalbumin or house dust mite (HDM) extract on top of this inflammation, however, would not be very instructive in our opinion because of this massive background inflammation even if *Stim1/2^{FOXP3}* mice might show further exacerbation of pulmonary inflammation. We would like to point out that historically the adoptive transfer of Treg cells (together with effector or naive T cells) into lymphopenic *Rag1^{-/-}* host mice has been the most common and accepted method to test Treg function *in vivo*. The advantage of this approach is that the numbers of transferred T cells are known and can be titrated and are not perturbed by host T cells. By contrast, there are no well-established T cell transfer models to specifically test Treg function in asthma.

3. For a better comparison with the human patients harbouring STIM1 LOF mutations, why did the authors not work with only STIM1 deficient Tregs? **Response:** For the purposes of this manuscript, we wanted to completely delete SOCE to evaluate its role in Treg function and differentiation. This can best be achieved by deletion both *Stim1* and *Stim2* genes, rather than just one of them. In mice, STIM1 and STIM2 have somewhat redundant functions as is apparent, for instance, from reduced Treg numbers and splenomegaly & lymphadenopathy in *Stim1/2 Cd4Cre* mice, but not *Stim1 Cd4Cre* or *Stim2 Cd4Cre* mice (Oh-hora 2008, PMID: 18327260). Similarly, antiviral immunity to LCMV is impaired only in *Stim1/2 Cd4Cre* mice, but not *Stim1 Cd4Cre* or *Stim2 Cd4Cre* mice (Shaw 2014, PMID: 25157823). By contrast, human patients with LOF mutations in STIM1 lack SOCE completely and develop autoimmunity (AIHA), may have reduced Treg numbers in their blood, and

are immunodeficient with increased susceptibility to viral infections. These findings suggest that deletion of STIM1 alone in humans results in a more pronounced defect than in mice (**Table S1**). We therefore reasoned that *Stim1/2*-deficient mice represent a good model for STIM1-deficient patients.

Why was the transcriptome analysis not performed with human patient and control Tregs? **Response:** All data in this manuscript are generated using Treg cells of WT and *Stim1/2^{FOXP3}* mice and we therefore also conducted the transcriptome analysis in mouse Treg cells to ensure that we could directly compare the gene expression signatures with Treg phenotypes and function. We agree that a transcriptome analysis in human Treg cells lacking SOCE would be very interesting and we are considering to do it as part of a follow-up study once new patients with STIM1 LOF mutation become available. PBMC of immunodeficient patients are rare and not always available for analysis (as many patients received curative bone marrow transplantation). Another factor that complicates transcriptome analysis in human STIM1-deficient patients is the difficulty to acquire appropriate age- and sex-matched controls. Using a sensitive method such as RNA-seq will result in many unspecific effects due to differences in age, sex, genetic background or prior infections of the individuals tested that may be unrelated to STIM1 deficiency.

4. In Figure S3B, the control of SOCE measurements in the Tcon controls isolated in S3A is missing as was done in Figure 1B. Minor: In both figures the time before readdition of Calcium is too short, as the intracellular stores are not completely depleted. **Response:** To respond to the reviewer's request, we have repeated the Ca^{2+} measurements in T cells from female hemizygous *Stim1/2 Foxp3-YFP^{Cre}* mice to compare SOCE levels in Treg cells and Tcon cells. We find that SOCE is almost completely abolished in *Stim1/2*-deficient YFP⁺ Treg cells compared to YFP⁻ Treg cells of *Stim1/2^{FOXP3}* mice (**revised Figure S2b**). SOCE levels in Tcon cells were intact and higher than in YFP⁻ Treg cells consistent with earlier findings (Gavin et al. 2002 Nature Immun, PMID: 11740498; Yan et al. 2015 PNAS, PMID:26627244). As requested by the reviewer, we have also added extracellular Ca^{2+} at a later time point to allow for full ER store depletion.

5. In Figure 1D the statistics are missing, the authors state that the populations are not shifted but the presented numbers show changes ranging from 12 – 25%. **Response:** We have added the requested statistical analysis of data in **revised Figure 1d**. There is a significant reduction in naive Treg cells in the spleen, but not in LNs, of *Stim1/2^{FOXP3}* mice.

6. On page 5, the authors state that there is not difference in Tbet or ROR γ T expression (data not shown), however, this data is very relevant considering the observed increase in IFN γ and IL17 expression and should therefore be presented. **Response:** We agree that these are important data and have added them to **revised Figure S1g**. We show that T-bet and ROR γ t expression is largely comparable in CD4⁺ Foxp3⁻ effector T cells from WT and *Stim1/2^{FOXP3}* mice. Only a moderate (although significant) increase in T-bet expression in T cells in LNs was observed, which is however much smaller than the increase in GATA3 (**Figure 2c**). These findings are consistent with the only 1.5-fold and 2.4-fold increase in IL-17 and IFN γ , respectively, in effector T cells from *Stim1/2^{FOXP3}* mice compared to the more pronounced increase in GATA3 and the Th2 cytokines IL-4 and IL-5.

7. No statistics are given in Figure 3G to show how significant the changes in Treg specific genes are. **Response:** We have added these data to **revised Figure 3g**. We show that expression of all genes (CD39, FolR4, Ox40, PD-1, ST2, ICOS, KLRG1, Nrp1, CD103) is significantly reduced in Treg cells of *Stim1/2^{FOXP3}* compared to WT mice.

Minor: Page 4 bottom typo: by -> but, same sentence contains wrong Figure reference (Figure 1E should be 1F). In Figure 2C the label colors in CD4+/FoxP3- cells are switched (compare numbers with bar graph in right hand panel). **Response:** We thank the reviewer for pointing out these mistakes, which have been corrected in the revised manuscript.